# Design and optimization of soft finger actuators for rehabilitation applications: A combined finite element and neural network approach

**Mahmoud Elsamanty**[1]*, **Karim Badr**[1,2], **Basem Akl**[1], **Abdelkader Ibrahim**[1], **Hongbo Yang**[3,4], **Kai Guo**[3,4]*, **Mostafa Orban**[1,3,4]

1 Mechanical Department, Faculty of Engineering at Shoubra, Benha University, Cairo, Egypt,
2 Mechatronics Engineering Department at Modern University for Technology and Information-MTI, Cairo, Egypt, 3 School of Biomedical Engineering (Suzhou), Division of Life Sciences and Medicine, University of Science and Technology of China, Hefei, China, 4 Suzhou Institute of Biomedical Engineering and Technology, Chinese Academy of Sciences, Suzhou, China

* mahmoud.alsamanty@feng.bu.edu.eg (ME); guok@sibet.ac.cn (KG)

**Data availability statement:** All relevant data are within the manuscript and its Supporting information files.

## Abstract

This study presents a comprehensive analysis of soft finger actuators using finite element modeling to assess their performance in various structural configurations. By conducting detailed numerical simulations, we explore how variations in structural parameters influence the bending angle, thereby guiding iterative design improvements. Specifically, the research examines the impact of critical design factors, such as the number of bellows, actuator height, surrounding thickness, and foot thickness, on the bending behavior of soft actuators. The objective is to optimize these actuators for use in rehabilitation training gloves, where precise motion control is para-mount. Our findings reveal that increasing both the height and the number of bellows significantly enhances the achievable bending angle, facilitating more effective rehabilitation exercises. In contrast, greater foot and surrounding thicknesses exhibit a restrictive effect on bending, underscoring the need to carefully consider these parameters in design processes. These insights are instrumental in formulating design guidelines that aim to optimize actuator performance in therapeutic applications. Crucially, the manuscript presents a rigorous comparison between the experimental results and simulation results, demonstrating a high degree of concordance that validates the FEM approach and the predictions of the neural networks. This close match between the observed and predicted data not only confirms the reliability of the simulations, but also enhances the credibility of the design recommendations for rehabilitation applications. Furthermore, the study uses artificial neural networks to predict bending angles with high precision. With a residual variance of just 0. 74% and an explained variance of 99. 26%, the neural network model demonstrates exceptional predictive capacity, highlighting its potential as a tool for further refinement of the design and optimization of the performance of soft actuators.

**Funding:** This research was funded by the Project of the State Administration of Foreign Experts Affairs (H20240225) (ME); the Chinese Academy of Sciences President's International Fellowship Initiative (2024VBB0010) (ME); the National Key R&D Program China (2023YFB4706200) (KG); Pilot Projects for Fundamental Research in Suzhou (SSD2023014) (KG); the Science and Technology Development Plan Project of Jilin Province (20240305049YY) (KG); the Natural Science Foundation Project (ZR2022QH214) (KG); and the Chongqing Natural Science Foundation Project (2024NSCQ-MSX0007) (KG). The funders had no role in the study design, data collection and analysis, decision to publish, or preparation of the manuscript.

**Competing interests:** The authors have declared that no competing interests exist.

This research not only advances our understanding of soft actuator mechanics, but also contributes to the development of more effective rehabilitation technologies.

## Introduction

Stroke is recognized as one of the leading causes of adult labor loss, significantly affecting the quality of life of individuals. Following a stroke, patients often experience mobility impairments, such as unilateral paralysis of the arm, leg, and face [1,2]. Hemiplegia, characterized by reduced motor function in the upper extremities, especially the hands, severely limits the ability of patients to perform activities of daily living (ADLs). The hand, being a distal component of the body, poses the greatest challenge in upper extremity rehabilitation [3,4]. Consequently, the degree of recovery from hand function serves as an important metric to assess overall upper limb rehabilitation progress in movement disorders. Rehabilitation robots have been identified as effective tools in rehabilitation training. These devices not only facilitate the recovery of motor function in patients' limbs but also substantially reduce the workload of rehabilitation therapists [5,6]. In recent years, numerous therapeutic methods have been developed to help restore upper limb movement after stroke. Among these, soft actuators have emerged as promising components due to their unique properties [7].

Globally, low- and middle-income countries account for 70% of strokes and 87% of stroke-related deaths [8]. In the last forty years, the incidence of stroke in low- and middle-income countries has increased by more than 100%. In contrast, during the same period, high-income countries have seen a 42% reduction in stroke incidence [9,10]. On average, strokes occur 15 years earlier in low- and middle-income countries and result in higher mortality rates compared to high-income countries. These events predominantly impact individuals during the most productive years of their lives [11].

Soft robotics has progressed rapidly in recent years, driven by its wide range of potential applications [12–16]. For example, using a hard gripper to hold smooth and delicate objects can easily cause deformation [17–20]. In contrast, soft grippers allow for safer and more precise handling of such objects [21–23]. Soft actuators are increasingly used for their compliance and similarity to human muscles. They are considered safe for human-robot interfaces, cost-effective, lightweight and capable of generating high power levels with significant mechanical output [24]. It is suggested that simple theoretical simulations be employed to understand the geometry and mechanisms of soft-finger actuators (SFA) [25–27]. The design of soft pneumatic finger actuators can therefore be based on this prototype, offering insight into their potential applications. The operating principles of the actuator with respect to the pressure air and the bending angles are defined. The actuator utilizes a differential pressure actuation mechanism to achieve bending motion [28,29]. Upon application of external pressure, volumetric expansion occurs within the internal chambers. This expansion is resisted by a rigid and incompressible footer that remains unaffected by applied pressure. The resulting asymmetry in expansion between the chambers and the footer induces a bending moment that causes the actuator to deflect. This innovative approach highlights the potential of soft actuators in the advancement of therapeutic technologies and the improvement of the effectiveness of rehabilitation interventions.

The performance of soft actuators is crucial for the effectiveness of rehabilitation training gloves, which led to extensive research efforts. The impact of design parameters such as the thickness of the underlying layer, the gaps in the chamber, and the thickness of the chamber wall on actuator performance has been examined, revealing influential patterns [30]. The introduction of the free bottom actuator (FBA) has been shown to enhance bending angles

and output force by eliminating constraints at the bottoms of the airway sections [31]. In addition, research on soft pneumatic actuators (SPAs) has demonstrated through finite element method (FEM) simulations that reducing wall thickness increases bending angles while varying chamber angles induce both bending and twisting motions, thus improving deformation and bending capabilities [32]. Despite these advances, challenges related to stiffness persist, highlighting the need for further investigation. The effects of chamber shape, number, and bottom cross-section dimensions on performance have been explored. However, ergonomic requirements for designing soft actuators in rehabilitation gloves remain incompletely addressed in current studies [33].

Neural networks, as computational models inspired by the biological brain, have emerged as foundational elements in artificial intelligence research. Through simulation, artificial versions of these networks are constructed using experimental data, serving as powerful tools to address application-specific challenges in modeling and control [34]. By emulating the learning processes of biological neural networks, these simulations bridge the gap between understanding the brain and developing intelligent machines capable of performing complex tasks [35,36]. Artificial neural networks (ANNs) offer a robust mathematical framework for understanding and simulating the intricate workings of the brain. Using mathematical equations, ANNs capture the essence of biological neurons and their interconnections, incorporating information flow and key processing parameters. This framework not only represents individual neurons and their connections, but also models the flow of information through the network, reflecting both the structure and characteristic behaviors of the brain [37]. In this study, a neural network approach was used to model the bending angle of the pneumatic soft actuator, and the results were validated by testing. An artificial neural network (ANN) was employed to predict actuator performance, with the primary objective of minimizing output error and achieving rapid convergence. The ANN was trained on a dataset from previous finite element simulations, using four input parameters: height, number of bellows, thickness surrounding the bellows, and foot thickness, with one output: bending angle.

Despite recent advances in soft actuation for manipulation and mobility, three limitations persist for hand rehabilitation: (i) many actuators achieve their reported bending or force output at ≥120–150 kPa (and up to 500 kPa), which burdens portable hardware and raises safety concerns for prolonged, home-based use; (ii) general-purpose designs often lack anatomical segmentation and joint conformity, increasing localized skin pressure and control complexity (e.g., multi-channel pressure sequencing); and (iii) predictive design tools that couple validated FEM with efficient, data-driven surrogates remain underutilized for rapid, low-pressure optimization. Addressing these gaps, this work develops a segmented, anatomically inspired soft finger actuator that delivers large, controllable bending at sub-100 kPa with a single pneumatic input, and establishes a combined FEM-ANN pipeline experimentally validated to accurately predict bending from four key geometric parameters (H, N, TSB, TOF). A rehabilitation-centric comparison with representative actuator families has been provided, normalized by operating pressure, to contextualize the design advantages in terms of safety, ergonomics, and simplicity of control.

This paper investigates the impact of parameters, including the number of bellows, the height of the bellows, the thickness of the surrounding material and the thickness of the foot, on the bending angle of soft actuators. Understanding how these parameters influence bending is crucial for optimizing the performance of soft actuators in various applications, particularly in rehabilitation training gloves and prosthetic hands. Where, precise and controlled motion is essential for effective rehabilitation. This research aims to identify optimal design configurations that maximize bending capabilities, thereby enhancing the functionality and usability of soft actuators in therapeutic settings.

## System modeling

### Modeling of soft bionic hand

Soft robotic bionic hands mark a significant advancement in prosthetics and robotics, moving away from traditional rigid designs to adopt more flexible and adaptive systems. These innovative devices leverage soft materials combined with advanced robotics to closely replicate the intricate dexterity and sensitivity of the human hand [38,39]. The field of soft robotics is defined by its use of compliant and deformable structures that promote natural interactions with the environment and improve grasping capabilities [40]. By incorporating the biomechanics of the human hand, bionic hands are equipped with sophisticated sensors and actuators, enhancing the user's sense of touch and control [41]. This integration of soft robotics and bionics overcomes the limitations of conventional prosthetics by creating an intuitive and comfortable interface between the user and the device, potentially improving the quality of life of individuals with limb loss by helping them regain functional independence and easily perform everyday tasks.

This study introduces a robotic hand designed to emulate the natural movements of the human hand, considering variations in finger length and proportions, as illustrated in Fig 1. The structure of the human hand features a multi-joint arrangement of the fingers, crucial for the execution of complex movements [42]. Including the thumb, typical fingers consist of three joints: the metacarpophalangeal joint (MCP) at the base, the proximal interphalangeal joint (PIP) in the middle, and the distal interphalangeal joint (DIP) at the tip. Each joint plays a vital role in providing the hand with flexibility and dexterity. The MCP joint, for example, supports a wide range of movements such as flexion, extension, and lateral motion, while the PIP and DIP joints mainly aid in bending and straightening the fingers, often synchronously.

The MCP joint functions independently with a unique air inlet/outlet channel, whereas the PIP and DIP joints are interconnected and share a single channel. Both the MCP and PIP joints have an equal number of air chambers, but the DIP joint has one fewer chamber. These chambers are semicircular in shape. The design of soft-driven joints in the bionic fingers integrates both fast pneumatic structures (FPN) and pneuNet technologies. The FPN structures utilize air-filled elements such as membranes, inflatable tubes, or chambers organized in a grid pattern. The effectiveness of FPN depends on the response to external air pressure applied to the inner walls of these chambers, causing them to expand and deform, thereby increasing the length of the drive layer [43,44].

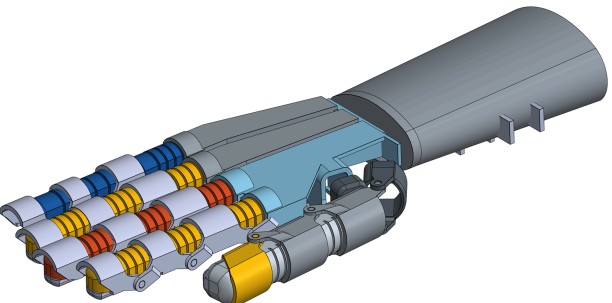

**Fig 1. 3D model of the soft finger actuator fabricated from silicone rubber using a 3D printing process.** This manufacturing approach allows for customization and rapid prototyping.

This sophisticated understanding of biomechanics and technological integration in bionic hands lays a solid foundation for the design of specialized actuators in rehabilitation devices. As we shift focus from modeling soft bionic hands to their specific application in rehabilitation, the next section will explore how these principles are adapted to develop soft finger actuators. These actuators not only mimic human anatomy, but also meet the therapeutic needs of individuals recovering from neurological injuries.

## Modeling of soft finger actuators

The primary goal of hand rehabilitation training devices is to help stroke patients regain hand function. Through repetitive motion exercises, these devices help patients gradually restore the motor functions of their fingers. For effective training, the design of the device must adhere to ergonomic principles [6]. Key hand movements include grasping, hooking, pinching, and dynamic manipulation, all of which involve bending, extending, abducting, and adducting finger joints. These movements are directly related to the structural characteristics of the finger, which consists of multiple bones. For example, the index finger is made up of the distal phalanx (DP), the middle phalanx (MP), and the proximal phalanx (PP), connected by joints and ligaments and controlled by tendons to facilitate joint flexion and extension [45].

The actuator is the main driving component of hand rehabilitation training devices, significantly influencing their rehabilitation effectiveness. Prioritizing safety and ease of use, these devices utilize pneumatic soft actuators as driving elements [46]. To meet ergonomic requirements, the soft actuators in this study are designed with a segmented structure, allowing each joint and phalanx to be independently controlled. The segmented joint-mimetic geometry is intentionally paired with a low-durometer silicone to concentrate volumetric expansion at the bending loci and achieve large angular excursions at sub-100 kPa. This material-architecture synergy reduces the burden of pneumatic hardware and enhances user comfort by maintaining a soft conformal interface with minimal localized pressure on the skin.

The geometry of the finger soft actuator plays a critical role in its effectiveness. The actuator in this paper adopts a three-part structure that mirrors the anatomy of the human finger [47,48], as shown in Figs 2 and 3. This design choice, with its carefully selected geometrical parameters, offers two key benefits: improved functionality and enhanced comfort, which are

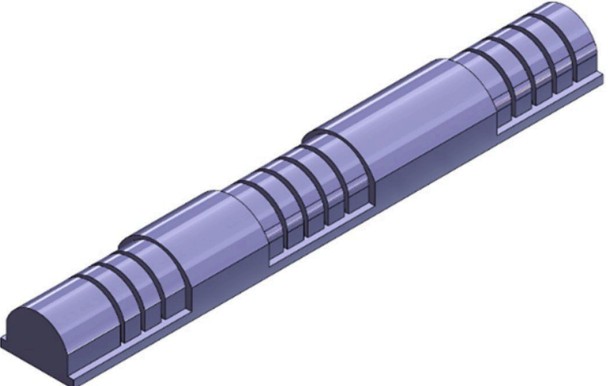

**Fig 2. 3D model of the soft finger actuator fabricated from silicone rubber using a 3D printing process.** This manufacturing approach allows for customization and rapid prototyping.

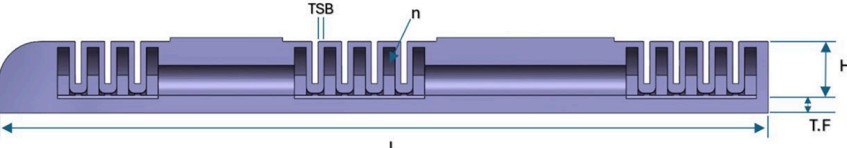

**Fig 3. Geometrical parameters of the soft finger actuator.** The design incorporates multiple air chambers (N), each with a defined wall thickness (TSB) and height (H). The overall length (L) and foot thickness (TF).

particularly relevant for rehabilitation applications. By mimicking the structure of the finger, the actuator conforms more naturally to the user's hand, potentially reducing pressure points and leading to a more comfortable user experience during therapy.

Parameter Selection Rationale for Rehabilitation Context. Three practical considerations bounded the geometric ranges investigated in this study: (i) rehabilitation-driven kinematics at sub-100 kPa, (ii) anatomical conformity to finger segment lengths, and (iii) fabrication and sealing reliability in monolithic, 3D-printed silicone. Preliminary screening (FEM and bench-top pilots) indicated that H below 10 mm produced limited bending and high sensitivity to wall tolerances, while H above 25 mm introduced local wrinkling or out-of-plane deformation unless wall thicknesses increased, which conflicted with our low-pressure objective. The number of bellows N was confined to 12–20 to distribute curvature over a 130 mm actuator length without reducing the pitch of the chamber below the printable limits or increasing the pneumatic resistance; N < 12 underperformed in angular excursion, and N > 20 compromised printability and sealing. The thickness of the surrounding wall TSB was limited to 1.0–1.75 mm to avoid leakage/buckling at the low end and excessive stiffness at the high end; the foot thickness TOF was set at 1.25-2.0 mm to balance the bending efficiency with the structural support and the stability of the interface. These empirically validated bounds defined the design-of-experiments grid used for FEM sweeps and ANN training, ensuring relevance to a portable and safe hand rehabilitation operation.

Although soft pneumatic actuators have been widely explored in robotics, existing designs often prioritize general-purpose functionality over specialized demands of hand rehabilitation. For example, PneuNets [16] and fiber-reinforced actuators [17] achieve high force or multidirectional bending but require pressures greater than 150 kPa, increasing the complexity of the system and the safety risks to patients. Similarly, McKibben muscles [18], though robust, are limited to linear motion and lack the anatomical fidelity required for finger joints. In contrast, the actuator proposed in this work integrates a segmented, ergonomic structure with 3D-printed monolithic fabrication, enabling patient-specific customization and safe operation at sub-100 kPa pressures. To contextualize these advances, Table 1 compares our design with the latest soft actuators, highlighting its superior bending performance, ergonomic compliance, and rehabilitation-centric optimization. Key metrics include bending angle, pressure requirements, and fabrication complexity, underscoring its potential as a tailored solution for post-stroke hand therapy.

## Comparative studies with existing soft actuators

Comparative Positioning within State-of-the-Art Soft Actuators. To contextualize the proposed actuator within the broader field, we contrast its performance and design attributes against three representative families: PneuNets, fiber-reinforced bending actuators, and McKibben muscles. As summarized in Table 1, our actuator achieves a maximum bending angle of 115° at 90 kPa, while operating within a sub-100 kPa envelope that is attractive for

**Table 1. Comparison of proposed soft finger actuator with state-of-the-art technologies.**

| Parameter | This Work | PneuNets [16] | Fiber-Reinforced [17] | McKibben [18] |
|---|---|---|---|---|
| Max Bending Angle | 115° at 90 kPa | 85° at 120 kPa | 70° at 150 kPa | Linear contraction (30%) |
| Force Output | 1.8 N (axial) | 2.5 N (multi-chamber) | 3.2 (high stiffness) | 15 N (tensile) |
| Response Time | 0.8 s (0–90 kPa) | 1.2 s (0–120 kPa) | 2.0 s (0–150 kPa) | 0.5 s (fast contraction) |
| Fabrication | *3D-printed, monolithic* | Multi-layer molding | Manual fiber wrapping | Braided sleeve + bladder |
| Pressure Range | 0–90 kPa | 0–200 kPa | 0–250 kPa | 0–500 kPa |
| Ergonomics | **Segmented, finger-joint mimicry** | Flat, multi-chamber | Rigid-soft hybrid | Bulky, non-anthropomorphic |
| Control Complexity | Low (single input) | Medium (multi-channel) | High (pressure sequencing) | Medium (pressure modulation) |

rehabilitation safety and system simplicity. PneuNets often require up to 120–200 kPa to reach comparable deformations and involve multi-layer molding, whereas fiber-reinforced actuators can deliver high forces but typically at ≥150 kPa with increased control complexity due to pressure sequencing and anisotropic reinforcements. McKibben muscles provide large linear forces but are not inherently joint-conformal and generally demand higher operating pressures (≥300–500 kPa). In contrast, our segmented, anatomically inspired architecture combines large bending at lower pressure, monolithic fabrication for rapid patient-specific customization, and low control complexity (single pneumatic input), thereby addressing clinically relevant constraints for hand rehabilitation.

Although soft pneumatic actuators have been widely explored in robotics, existing designs often prioritize general-purpose functionality over the specialized demands of hand rehabilitation. For context, we compare the proposed segmented, monolithic actuator against three representative families: PneuNets, fiber-reinforced bending actuators, and McKibben muscles. Table 1 summarizes key metrics under comparable operating conditions. Our actuator achieves a maximum bending angle of 115° at 90 kPa, emphasizing efficient deformation within a sub-100 kPa envelope that aligns with clinical safety and portable pneumatic hardware. PneuNet actuators commonly require 120–200 kPa for similar angular excursions and depend on multi-layer molding. Fiber-reinforced actuators can deliver higher directional force but typically operate at ≥150 kPa and involve increased control and fabrication complexity due to anisotropic reinforcement. McKibben muscles provide high linear force but are not inherently joint-conformal and commonly require 300–500 kPa. The proposed design combines: (i) joint-mimetic segmentation that concentrates volumetric expansion at bending loci, (ii) tuned wall (TSB) and foot (TOF) thicknesses that avoid unnecessary stiffness, and (iii) monolithic 3D-printed fabrication that preserves chamber fidelity and minimizes leakage. These factors jointly enable higher bending per unit pressure with low control complexity (single pneumatic input), enhancing suitability for hand rehabilitation.

Standardized Comparison and Normalization. To reduce bias across platforms, we emphasize comparisons within overlapping pressure ranges and normalize bending efficiency by reporting angle at sub-100 kPa, where available. Although exact actuator geometries differ, the segmented architecture of the proposed design and low-durometer silicone consistently increase bending per unit pressure relative to fiber-reinforced and chambered baselines operated above 120–150 kPa. Where direct force normalization is limited by architectural differences (for example, McKibben linear contraction), we contextualize rehabilitative relevance through ergonomic conformity, simplicity of control and safety envelope, which are critical determinants for clinical deployment.

Interpreting the Comparison. The observed performance advantages at lower pressure are attributable to: (i) segmented, joint-mimetic geometry that concentrates volumetric expansion where bending moments are maximized; (ii) the tuned balance between wall

thickness (TSB) and foot thickness (TOF), which reduces unnecessary stiffness without compromising structural integrity; and (iii) monolithic 3D printing that preserves chamber fidelity and minimizes leakage at interfaces. These design choices jointly enable higher bending per unit pressure, which is advantageous in clinical environments that favor compact, low-pressure pneumatic hardware.

Several geometric dimensions characterize the soft pneumatic actuator used in this investigation. These parameters include the number of air chambers (N), Thickness of the Surrounding wall Bellows (TSB), Thickness Of Foot (TOF), total length (L), and air chamber height (H). These dimensions are summarized in Table 2.

## Finite element analysis

While analytical models often struggle to capture the nuances of soft actuator behavior due to their intricate geometries, non-linear material properties, and the influence of air compressibility, the finite element method (FEM) has emerged as a powerful tool in the field of soft robotics. The ability of FEM to discretize complex geometries into smaller, more manageable elements allows it to accurately simulate the large deformations inherent to soft actuator operation, encompassing both the deformation of the material itself and the internal pressure changes during inflation [48–51]. This capability translates into several significant advantages. First, FEM enables researchers to predict the performance of soft actuators under a wide range of input conditions, including varying pressures, external loads, and material properties [52,53]. This predictive power translates into a more streamlined and cost-effective design process, as it reduces the reliance on costly and time-consuming physical prototypes. Beyond predicting overall actuator behavior, FEM provides critical insights into the internal mechanics of soft actuators. By mapping stress concentrations and strain distributions within the actuator's structure, FEM allows for a more accurate assessment of fatigue performance [54]. This is particularly crucial for applications where actuators un-dergo repeated cycles of deformation, as it allows designers to identify potential failure points and optimize the actuator's lifespan. Furthermore, FEM excels in its ability to handle non-linear contact, which are particularly prevalent in soft robotic systems that interact with complex and unstructured environments [55]. Whether it is a soft gripper that grasps an object or a soft robot that navigates through confined spaces, FEM can accurately model the forces and deformations arising from these interactions, leading to more robust and reliable designs.

Low-Pressure Operating Envelope and DOE Grid. Consistent with our rehabilitation-centric constraint, all FEM simulations were executed at 0.09 MPa (90 kPa). The DOE grid spanned $H \in (10, 15, 20, 25)$ mm, $N \in (12, 14, 16, 18, 20)$, $TSB \in (1.0, 1.25, 1.5, 1.75)$ mm, and $TOF \in (1.25, 1.5, 1.75, 2.0)$ mm, reflecting the validated fabrication window and the anatomical form factor of a wearable glove. This grid ensured sufficient coverage of monotonic and

**Table 2. List of variables and their descriptions used as input parameters during the simulation process of the soft finger actuator.**

| Parameter | Value |
|---|---|
| N | 12.14, 16, 18, 20 |
| H | (10, 15, 20, 25) mm |
| TSB | (1, 1.25, 1.25, 1.75) mm |
| TOF | (1.25, 1.5, 1.75, 2) mm |
| Pressure | 90 KPa |
| Width | 18 mm |
| Length | 130 mm |

interaction effects observed in preliminary pilots, while avoiding configurations known to cause leakage, footer distortion, or excessive stiffness at ≤90 kPa.

Consistency with and Differentiation from Prior Actuators. Our parametric trends—namely that increasing chamber height (H) and the number of chambers (N) increase bending, whereas increasing surrounding thickness (TSB) and foot thickness (TOF) decrease bending—are consistent with foundational findings in soft pneumatic architectures (e.g., PneuNet-style chambers and fiber-reinforced geometries). However, the segmented, finger-joint-centric configuration herein yields these gains at lower operating pressures, enabling clinically desirable deformation without resorting to high-pressure infrastructure or complex reinforcement schemes. This represents a distinct positioning relative to general-purpose soft actuators primarily optimized for maximal force or workspace at higher pressures.

## Hyper-elastic material characterization

To account for the anisotropic properties of silicone rubber material, which significantly influences its overall characteristics, a standard tension sample was fabricated and subjected to uniaxial tensile testing. The testing procedure adhered to ISO 37 standards [50] and was carried out using a universal testing machine. The geometry of the specimen is depicted in Fig 4. Subsequently, a finite element model (FEM) was developed to predict the bending behavior and mechanical output of the soft actuator. Although various commercial software packages are available for FEM analysis in soft robotics, including Abaqus, ANSYS, COMSOL, and Marc, this study used ANSYS Workbench 2022 R2. The model used static structural analysis to simulate the actuator response to applied loads.

Material Selection Rationale for Rehabilitation Context. We selected a platinum-cured silicone rubber as the primary elastomer due to its favorable combination of skin-contact safety, tunable compliance, and stable hyperelastic response under large deformations. Medical-grade silicone formulations are widely used in dermal contact applications and can be certified according to ISO 10993 pathways, according to hygiene and safety requirements in clinical environments. From a mechanical point of view, the low-to-moderate Shore A hardness and low hysteresis of silicone allow large bending at sub-100 kPa with rapid elastic recovery, which is critical for comfortable and repetitive motion assistance. The advantages of the processing (room temperature curing, strong interlayer adhesion, and low interface permeability) facilitate the fabrication of monolithic chambers with a reduced risk of leakage, supporting the patient-specific customization central to rehabilitation. Alternative elastomers, such as TPUs and urethanes, offer higher tear resistance, but typically require increased

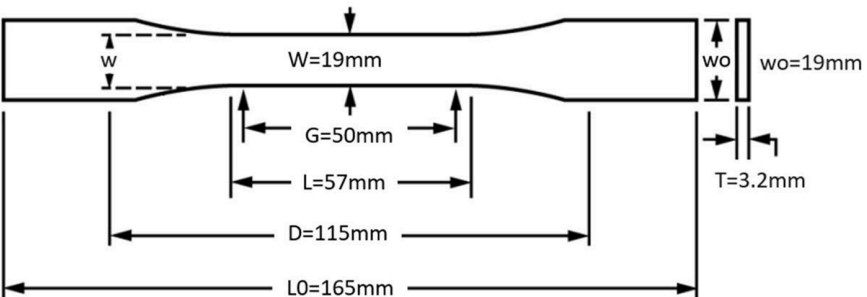

**Fig 4. Detailed dimensions of the test specimen used to characterize the mechanical properties of the silicone rubber material (ISO 37 compliant [28]).**

thickness or higher pressures to achieve similar bending, countering our low-pressure design objective and may reduce comfort. Fiber-reinforced laminates can increase force output but introduce stiffness anisotropy and fabrication complexity, reducing ergonomic compliance. Thus, the chosen silicone balances safety, compliance, manufacturability, and modeling fidelity, which collectively underpins the observed agreement between FEM predictions and experiments.

To accurately represent material behavior within the FEM, the experimental data from the uniaxial tensile tests was used to derive a hyperelastic material model as presented in the supplement data. The stress-strain curve obtained from the tests, illustrated in Fig 5, was fitted to a Yeoh 3rd-order hyperelastic model within ANSYS. This model effectively captures the nonlinear stress-strain relationship of the silicone rubber material. The fitting process yielded the following coefficients: $C_{10}$ = 3.061 MPa, $C_{20}$ = –637.651 kPa and $C_{30}$ = 98.623 kPa. This combined approach, which integrates both theoretical modeling and experimental validation, ensures the fidelity of the FEM and its ability to predict the performance of the soft actuator accurately.

**FEM mesh generation.** Given the hyperelastic nature of silicone rubber, accurately simulating the behavior of the soft actuator requires careful consideration of large deformations within the Finite Element Model (FEM). This study aims to understand how specific geometric parameters, namely, the actuator's height, thickness surrounding the bellows, foot thickness, and number of bellows, influence its bending angle. To achieve this, a robust meshing strategy is essential, starting with the selection of a tetrahedral element type due to its suitability for complex geometries like that of the soft actuator. A base size of 2 mm was chosen as a starting point, as shown in Fig 6.

However, uniform mesh density across the entire model can lead to an unnecessary high computational cost without significant gains in accuracy in less critical regions. Therefore, a slow sizing function was implemented to allow for a gradual transition in element size, with smaller elements concentrated in regions where stress concentrations or high strain gradients are expected, such as near the actuator's bending points or areas of contact. This approach ensures a balance between computational efficiency and accurate representation of the

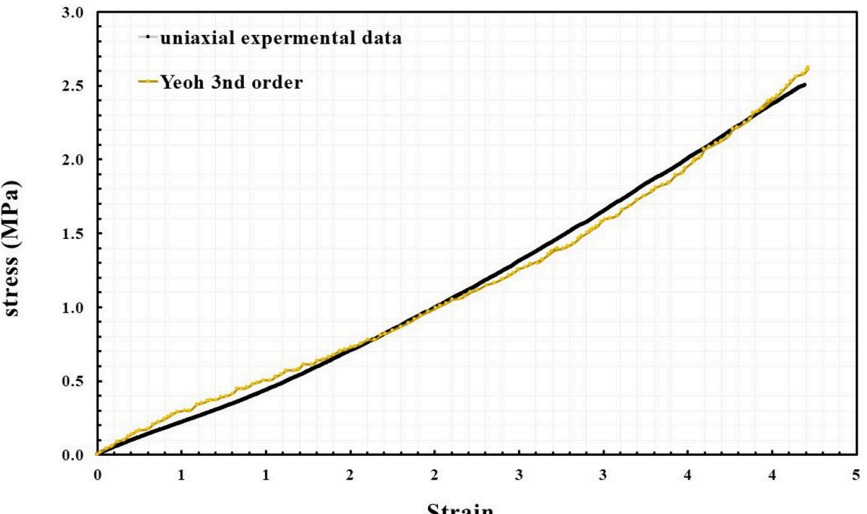

**Fig 5. Experimental stress-strain curve for the silicone rubber specimen, derived from uniaxial tensile testing.**

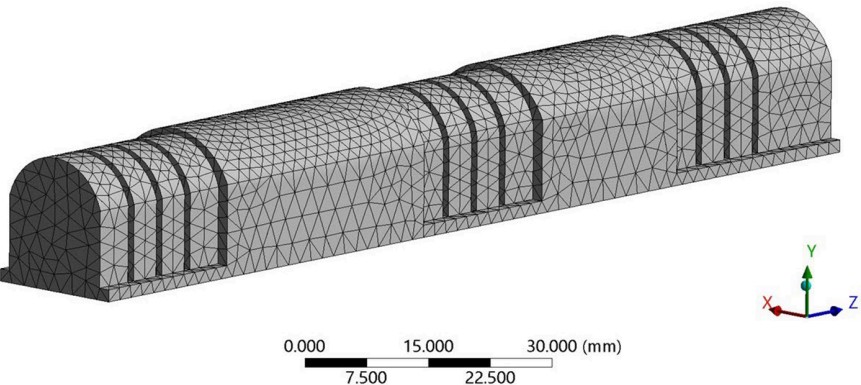

**Fig 6. Finite element mesh representation of the soft finger actuator.**

deformation. To further enhance the model's fidelity, particularly in capturing the intricate features of the actuator's design, local mesh refinement was implemented by setting the span angle at the element center to "Fine." This setting, which limits the maximum angle between adjacent element faces, results in a more refined mesh in areas of high curvature or geometric complexity, such as the bellows' convolutions or the transition zones between different actuator segments. This localized refinement leads to more accurate stress and strain calculations in these critical regions, ultimately improving the reliability of bending angle predictions.

To accurately simulate the actuator response to internal pressure, appropriate boundary conditions were carefully defined within the FEM environment. The actuator was conceptually modeled as a cantilever beam, a common simplification for structures anchored at one end and free to deflect at the other. This was implemented in the model by fully constraining the leftmost surface of the actuator, simulating a fixed support. The opposing end was left unconstrained, allowing free deformation under the influence of the applied pressure. To simulate the actuator's pneumatic actuation, a uniform pressure of 0.09 MPa was applied to the internal surfaces of the actuator model. Furthermore, the model incorporated the distinct material properties of the two primary layers of the actuator: the passive outer layer, as shown in Fig 7, and the active inner layer (highlighted in blue). This differentiation in material properties is crucial for accurately capturing the ac-tuator's bending behavior, as the

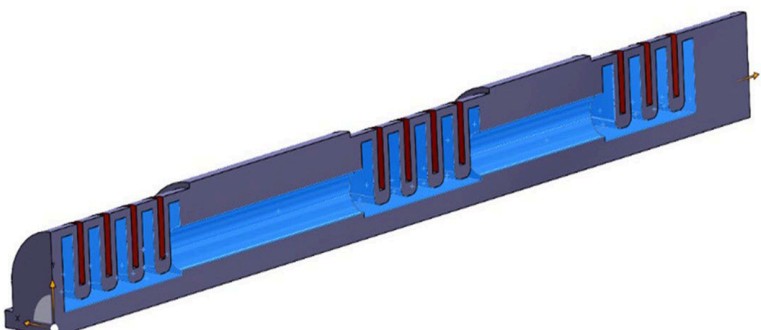

**Fig 7. Distinction between the passive outer layer (red) and the active inner layer (blue) within the ac-tuator model.**

interaction between the stiffer outer layer and the more deformable inner layer governs the overall mechanics of the system.

## FEM results discussions

This study undertakes a comprehensive examination of the influence of key geometric parameters on the bending response of a soft pneumatic actuator. Four parameters were selected for an in-depth analysis due to their anticipated impact on actuator performance: the number of bellows incorporated into the design, the individual height of each bellow, the thickness of the material surrounding the bellows, and the thickness of the actuator foot section. To isolate and quantify the effect of each parameter, a systematic analysis was performed. A series of finite element simulations were performed, systematically varying each parameter across a predetermined range while keeping the remaining three parameters constant. This resulted in a comprehensive dataset that captures the actuator's bending response to a wide spectrum of geometric configurations. To facilitate interpretation and visualization of this multi-parameter dataset, the results are presented as a series of comparative graphs. Each graph focuses on the interaction between two specific parameters chosen for their potential interdependency or significant influence on bending behavior. Within each graph, one parameter is systematically varied along the x-axis, while the second parameter assumes multiple discrete values, each represented by a distinct curve. This matrix-like visualization strategy allows for a nuanced understanding of how each parameter pair contributes to the overall bending response, revealing potential synergistic effects or trade-offs between design parameters. By systematically manipulating and analyzing these factors, this study aims to provide a comprehensive understanding of the design space and inform the optimization of soft actuators for specific applications.

### Influence of height on the bending angle

To isolate the impact of the bellow height on the soft pneumatic actuator (SPA) bending performance, a series of finite element simulations were performed, meticulously controlling for other geometric variables. Specifically, the number of bellows was kept constant at 12 in all simulations, ensuring that the observed effects were only attributable to changes in bellow height. Similarly, the thickness of the material surrounding the bellows was maintained at 1 mm, and the foot thickness at 1.25 mm, eliminating any potential influence from these parameters. Fig 8 presents the simulation results, visually depicting the relationship between the bellow height and the bending angle. The data reveal a clear and consistent trend: as bellow height increases, so does the actuator's bending angle. This positive correlation is evident across the entire tested range. For example, with a bellow height of 10 mm (represented by the blue line), the actuator achieves a bending angle of 78.94 degrees. Increasing the height to 15 mm (orange line) results in a notable increase in the bending angle to 94.43 degrees. This upward trend continues as the height is further incremented to 20 mm (green line), yielding a bending angle of 102.09 degrees. Finally, at the maximum tested height of 25 mm (red line), the actuator can reach its maximum reaching an angle of 106.55 degrees.

This monotonic relationship between bellow height and bending angle underscores the significant role this parameter plays in dictating the actuator's performance. The upward trend can be attributed to the mechanics of the below deformation. As the height of the bellow increases, the internal volume available for expansion under pressure also increases. This, coupled with the greater moment arm provided by a taller bellow, results in an amplified bending motion. These findings highlight the importance of carefully considering

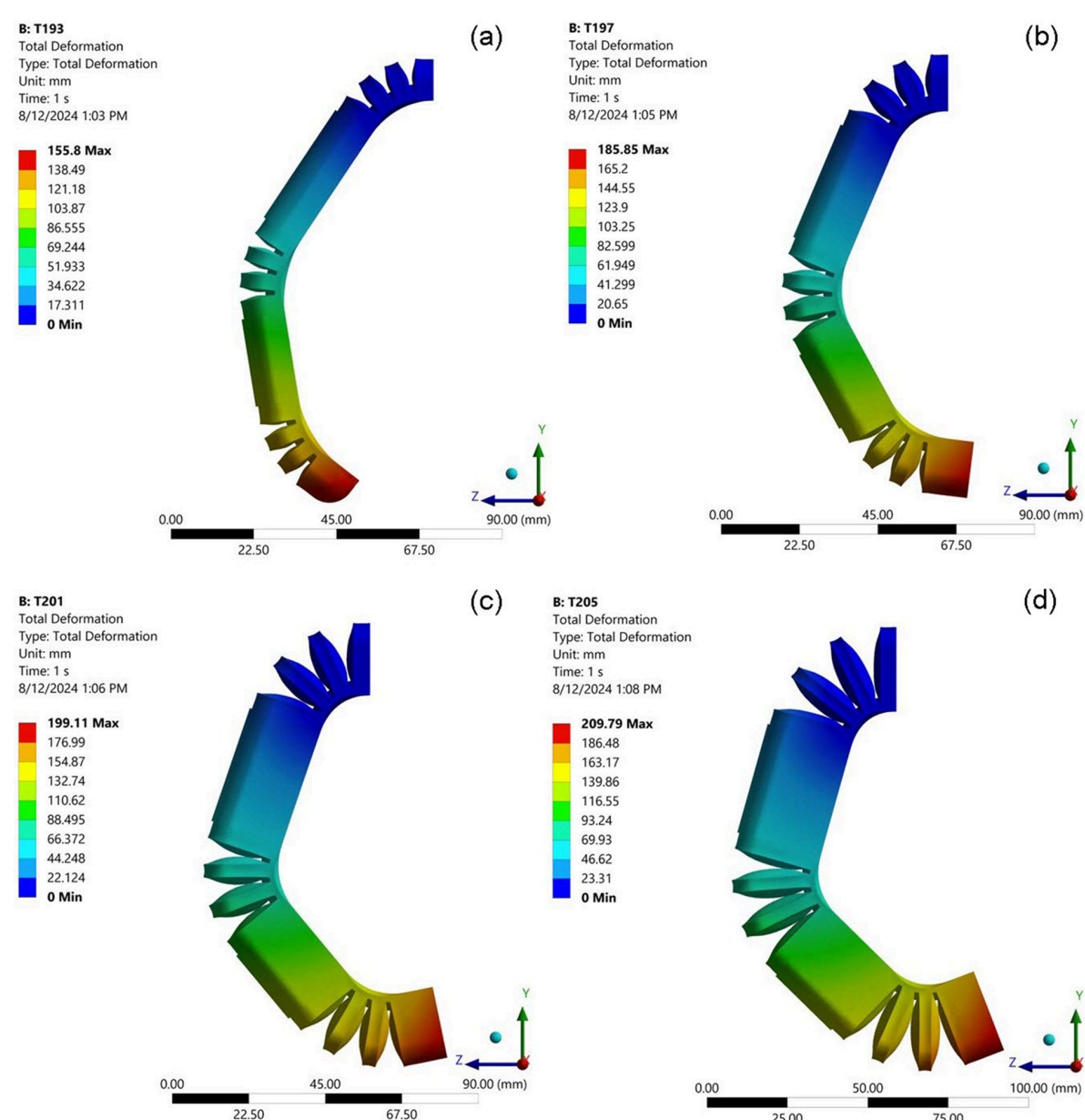

**Fig 8. Visualizations of the simulated bending behavior of the SPA under constant pressure, showcasing the effect of varying bellow heights (H).** (a) 10 mm; (b) 15 mm; (c) 20 mm; (d) 25 mm.

bellow height during the design phase, as even seemingly small adjustments to this parameter can lead to substantial changes in the actuator's bending capability.

## Influence the number of bellows on the bend angle

A comprehensive analysis was undertaken to elucidate the relationship between the number of bellows integrated into the soft pneumatic actuator (SPA) design and the resulting bending angle, a critical performance metric for this class of actuators. To isolate the effect of bellow count and mitigate the influence of other geometric parameters, a series of finite

element simulations were conducted with a fixed bellow height of 10 mm. This consistent height, represented by the blue line in all relevant data visualizations, ensured that any observed changes in bending behavior could be directly attributed to variations in the number of bellows. Fig 9 visually summarizes the simulation results, showcasing the SPA's bending response to an increasing number of bellows. The data reveal a clear and consistent trend: as the number of bellows increases, the actuator's bending angle exhibits a directly proportional increase. This positive correlation is evident across the entire tested range. For example, with 12 bellows, the SPA achieves a bending angle of 78.94 degrees. Incrementing the bellow count to 14 results in a substantial jump in the bending angle to 89.30 degrees. This upward trend persists as more bellows are incorporated into the design: 98.30 degrees at 16 bellows, 106.66 degrees at 18 bellows, approximately reaching a maximum observed bending angle of 115.03 degrees with 20 bellows.

This monotonic relationship between the number of bellows and the bending angle is not an isolated phenomenon. Consistent results are observed across a range of experimental conditions, as depicted in Figs 10, 11, 12, and 13. This repeatability across varying parameter sets highlights the robustness and generalization of this geometric dependency, following a fundamental mechanical principle governing the actuator's behavior. The observed trend can be attributed to the additive nature of the bottom deformation under pressure. Each individual bellow, when subjected to internal pressure, undergoes a degree of axial expansion. With a greater number of bellows connected in series, these individual deformations accumulate, resulting in a more pronounced overall bending motion. These findings underscore the importance of carefully considering the number of bellows during the design phase, as this parameter offers a direct and effective means of tailoring the bending characteristics of soft pneumatic actuators for specific applications and desired performance outcomes.

## Influence of surrounding bellows thickness on bending angle

Enhancing the height of the bellows and increasing the number of air bellows have both been shown to improve the bending angle of the soft pneumatic actuator (SPA). In contrast, the thickness of the surrounding bellows (TSB) has been identified as exerting a counteractive effect on this performance metric. To isolate and better understand the impact of TSB, a series of finite element method (FEM) simulations were executed, employing a consistent set of geometric parameters. Specifically, the height of each bellow was maintained at 15 mm, as indicated by the orange line in the corresponding figures, the number of bellows was fixed at 14, and the thickness at the base of each bellow (foot thickness) was kept constant at 1.25 mm. This rigorous control of variables allowed for a precise assessment of the influence of TSB on the bending behavior of the actuator. The results of these simulations are illustrated in Fig 14, which shows the effects of varying the TSB from 1 to 1.75 mm in increments of 0.25 mm. The results indicate that the maximum deformation observed was 193.49, providing critical insights into the relationship between TSB and the actuator's functional performance.

Fig 13 provides a visual representation of the simulation results, clearly showing the relationship between the TSB and the bending angle. The data reveal a consistent and inversely proportional trend: as TSB increases, the actuator's bending angle decreases. This negative correlation is observed across the entire range of TSB values tested. For example, with a TSB of 1 mm (as shown in Fig 13a), the SPA achieves a relatively large bending angle of 106.45 degrees. However, as TSB is incrementally increased, the actuator's ability to bend under pressure is progressively diminished. A TSB of 1.25 mm results in a bending angle of 97.40 degrees, while a further increase to 1.5 mm TSB yields a bending angle of 92.25 degrees. At the maximum tested TSB of 1.75 mm, the actuator exhibits its most constrained bending, achieving an angle of only 85.72 degrees. This monotonic decrease in the bending angle with

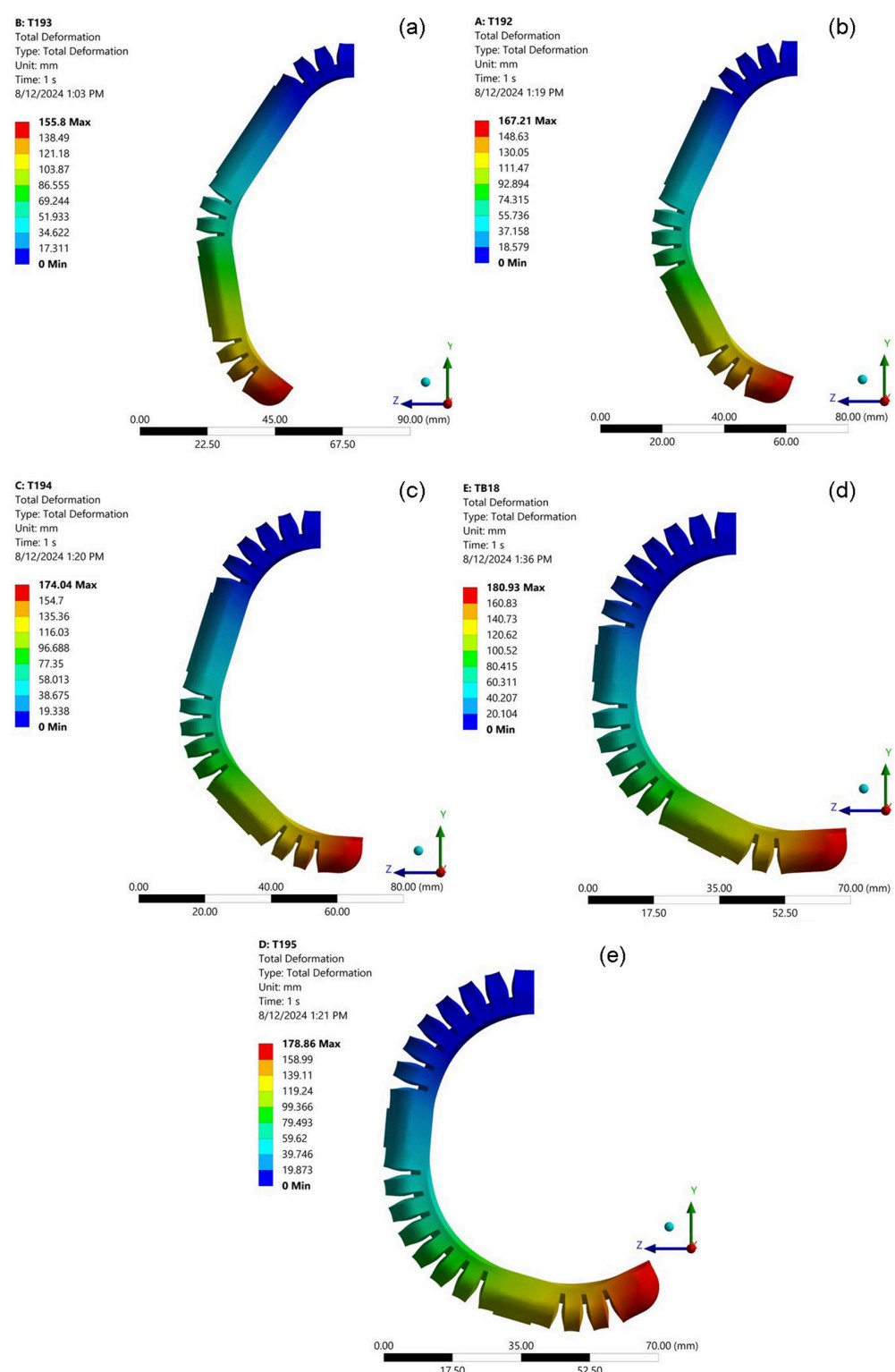

**Fig 9. Simulated deformed shapes of the SPA with varying numbers of bellows while maintaining a constant bellow height.** The bending angle increases with the bellow count: (a) 12; (b) 14; (c) 16; (d) 18; (e) 20.

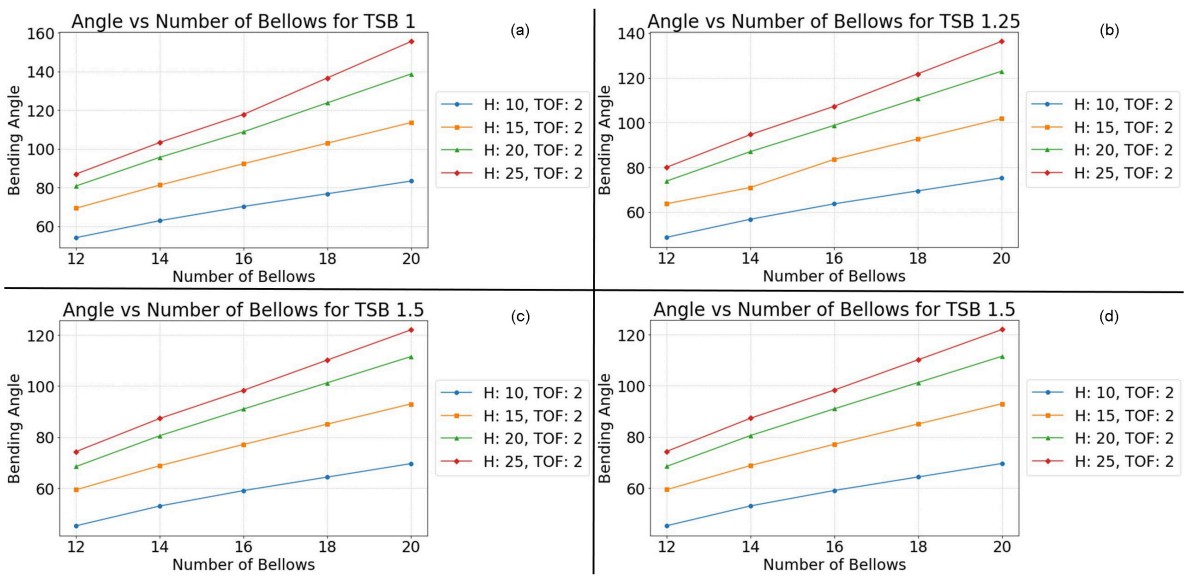

**Fig 10. Simulation results at Thickness of foot 2 mm.** (a) TSB = 1 mm; (b) TSB = 1.25 mm; (c) TSB = 1.5 mm; (d) TSB = 1.75 mm.

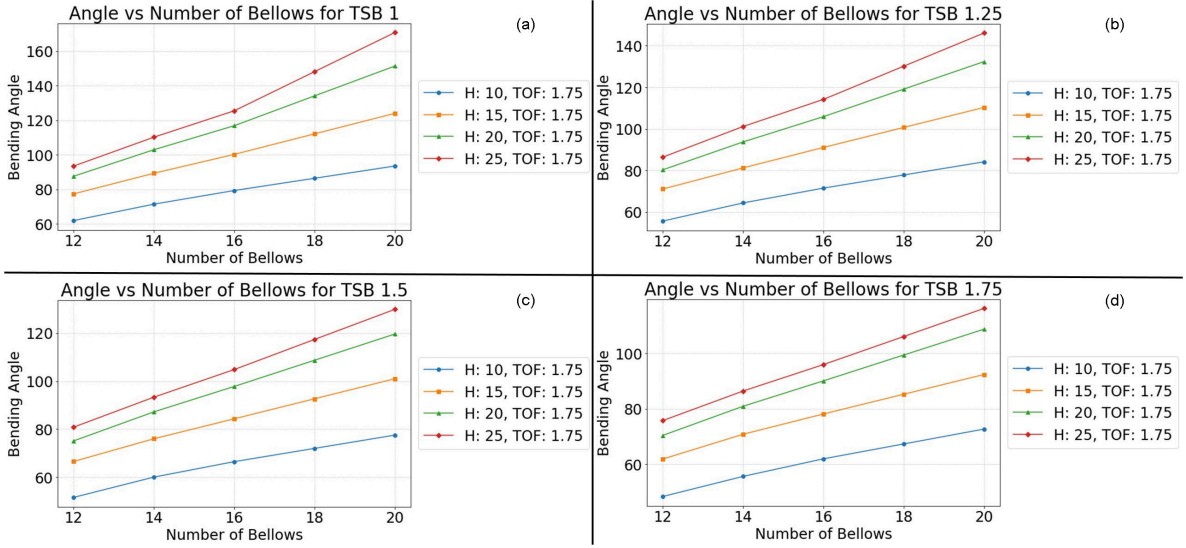

**Fig 11. Simulation results at Thickness of foot 1.75 mm.** (a) TSB = 1 mm; (b) TSB = 1.25 mm; (c) TSB = 1.5 mm; (d) TSB = 1.75 mm.

increasing TSB can be attributed to the fundamental mechanical properties of the actuator structure. The surrounding bellows, as their name suggests, provide structural support and define the actuator's overall geometry. A higher TSB translates to a higher concentration of material in these surround regions, effectively increasing the overall stiffness of the actuator. This enhanced stiffness requires a larger actuation force, in this case a higher internal pressure, to produce the same degree of deformation of the bending. The consistency of this trend across all data points, as evidenced in the presented figures, underscores the significant role of TSB as a design parameter for fine-tuning the stiffness and, consequently, the bending

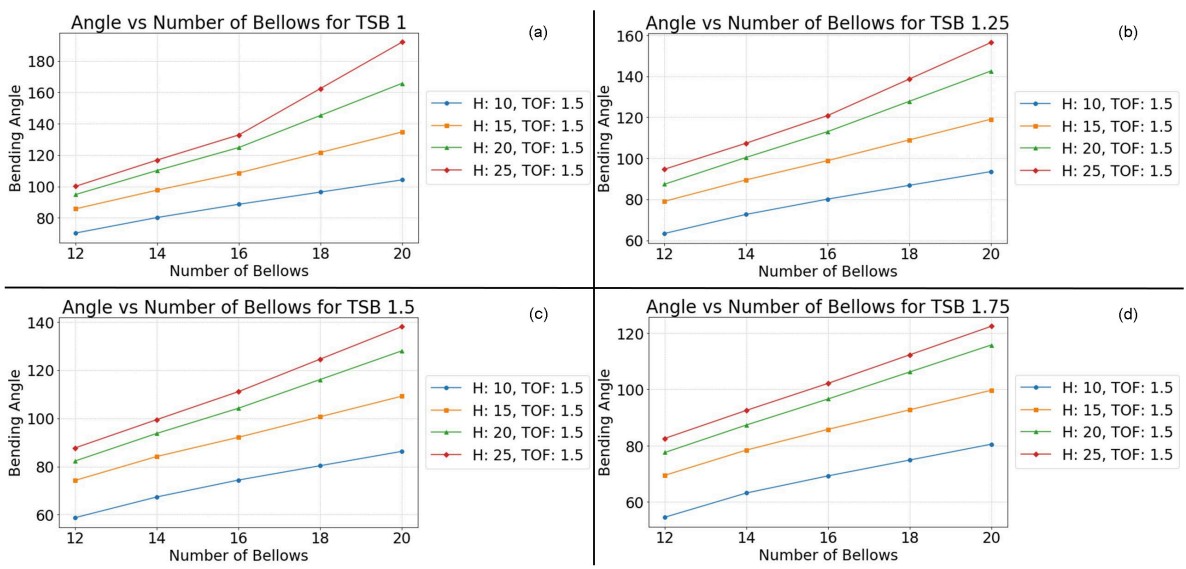

**Fig 12. Simulation results at Thickness of foot 1.5 mm.** (a) TSB = 1 mm; (b) TSB = 1.25 mm; (c) TSB = 1.5 mm; (d) TSB = 1.75 mm.

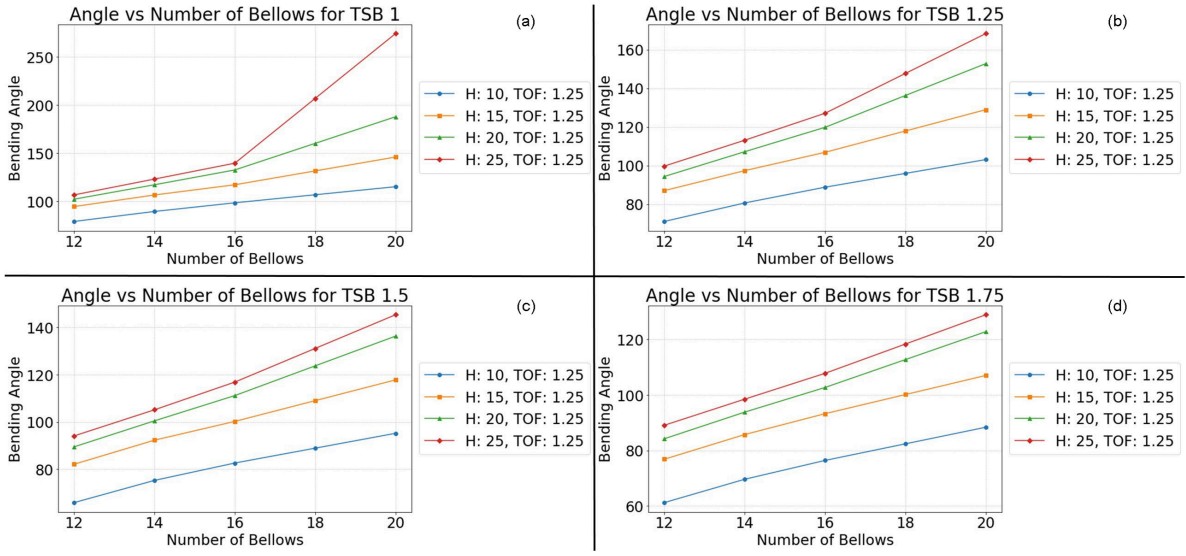

**Fig 13. Simulation results at Thickness of foot 1.25 mm.** (a) TSB = 1 mm; (b) TSB = 1.25 mm; (c) TSB = 1.5 mm; (d) TSB = 1.75 mm.

characteristics of the soft pneumatic actuator. By carefully considering and manipulating the TSB, designers can tailor the actuator's response to meet the specific force and displacement requirements of various applications.

## Influence of foot thicknes and bending response

The investigation of the geometric factors that influence the bending behavior of soft pneumatic actuators (SPA) has elucidated that both the thickness of the surrounding ribs (TSB) and the thickness of the foot actuator (TOF) are critical determinants. These parameters are essential for optimizing the design to enhance actuator performance. Fig 15 provides a

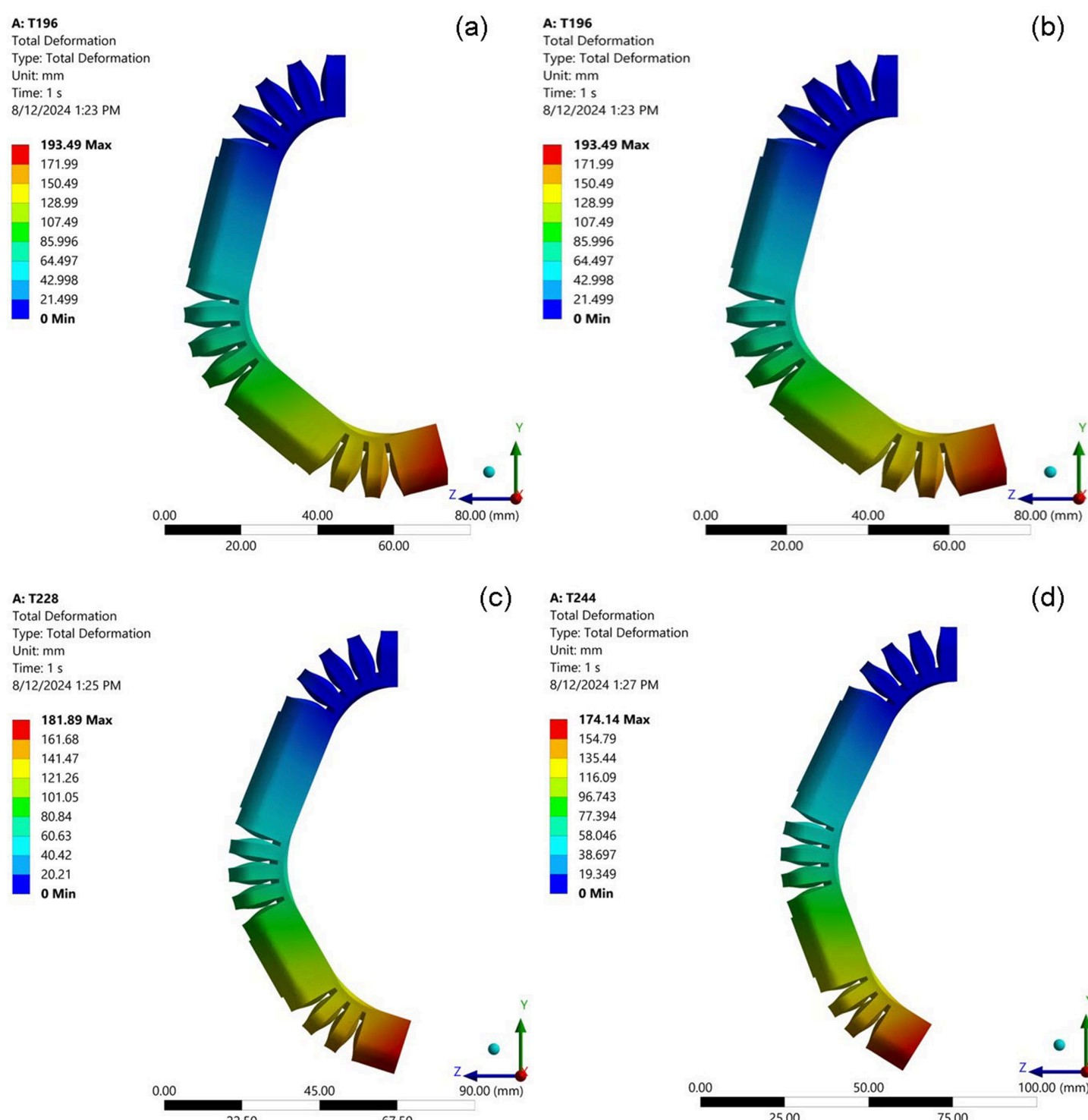

**Fig 14. Visualizations of the simulated bending behavior of the SPA under constant pressure, showcasing the effect of varying the thickness of surrounding bellows (TSB).** (a) TSB = 1 mm;(b) TSB = 1.25 mm;(c) TSB = 1.5 mm;(d) TSB = 1.75 mm.

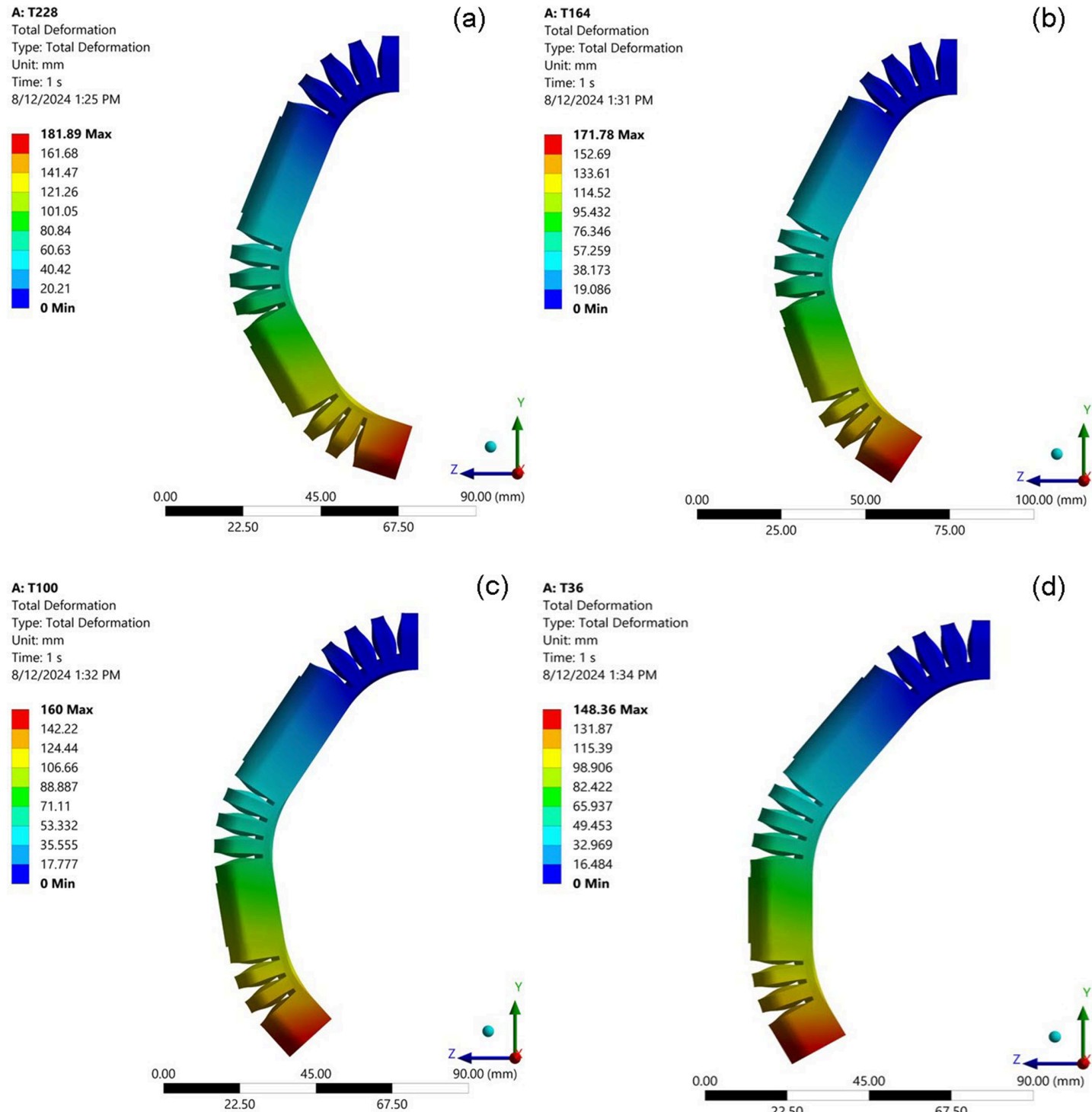

**Fig 15. Visualizations of the simulated bending behavior of the SPA under constant pressure, showcasing the effect of varying foot thickness.** (a) TOF = 1.25 mm; (b) TOF = 1.5 mm; (c) TOF = 1.75 mm;(d) TOF = 2 mm.

comprehensive analysis of the relationship between TOF and the bending angle of the SPAs, illustrating the data in an extensive range of TOF values. It is observed that an increase in the thickness of the actuator's foot correlates inversely with the bending angle. The underlying mechanical principles suggest that the structural support provided by the actuator foot, when increasing in thickness, increases the density of the material in this pivotal area, thus enhancing the stiffness of the SPA. Consequently, a greater internal force is required to achieve an equivalent degree of bending, resulting in a diminished bending angle for any given actuation pressure.

More detailed information is presented in Fig 13, which shows the impact of incremental increases in TOF on the bending angle. With a TOF set at 1.25 mm, the SPA exhibits a bending angle of 92.25 degrees. This angle decreases progressively with increases in TOF: a TOF of 1.5 mm results in a bending angle of 84.16 degrees, and increasing TOF to 1.75 mm results in a further reduction to 76.03 degrees. The minimum bending angle observed, 68.82 degrees, occurs at the maximum tested TOF of 2 mm. This consistently decreasing trend underscores the SPA's sensitivity to changes in foot thickness and emphasizes the necessity for precise parameter control.

To accurately assess the specific influence of TOF on bending performance, a series of finite element method (FEM) simulations were conducted. These simulations were performed with a strictly controlled set of geometric parameters to ensure that the observed effects were attributable solely to variations in TOF. The height of the bellows was consistently held at 15 mm, as indicated by the orange line in the accompanying figures, effectively isolating the effects of height variations. By keeping the number of bellows constant at 14 and maintaining a TSB of 1.5 mm, TOF was established as the primary variable under investigation.

The implications of these findings for the design and optimization of SPAs are significant. It is imperative for designers to recognize that although a thinner foot may facilitate larger bending angles, enhancing the actuator's displacement and motion profile capabilities, it may also compromise the structural integrity of the SPA. An excessively thin foot could lead to undesirable deformations or even structural failure under operating stresses. Thus, an optimal balance between bending performance and structural durability must be achieved. Design decisions regarding the thickness of the foot should be informed by a detailed understanding of the intended application's specific requirements and constraints, ensuring that the actuator not only meets the desired performance criteria but also maintains its structural integrity under practical conditions.

To ensure the reliability and clarity of the presented findings, a rigorous approach to parameter control was implemented across all simulations and their corresponding visualizations. This methodology allowed for the isolation and independent analysis of specific geometric factors influencing the bending behavior of the soft pneumatic actuator (SPA). To mitigate the potential for confounding effects arising from variations in foot thickness, this parameter was held constant within each figure while being systematically varied across different figures. Specifically, Fig 11 maintained a foot thickness of 2 mm, Fig 12 used 1.75 mm, Fig 12 employed 1.5 mm, and Fig 13 used 1.25 mm. This approach ensured that any observed differences in the bending behavior within a given figure could be attributed to the influence of the other geometric factors being investigated.

Within each Figs 10, 11, 12, and 13, the thickness of the surrounding bellows (TSB) was systematically modified to examine its impact on the performance of the SPA. This resulted in four distinct datasets per figure, each corresponding to a specific TSB value. Subfigures (a) employed a TSB of 1 mm, subfigures (b) used 1.25 mm, subfigures (c) utilized 1.75 mm, and subfigures (d) implemented 2 mm. By comparing the results across these subfigures, the influence of TSB on the bending angle, while controlling for other geometric factors, could be

effectively isolated and analyzed. To facilitate straightforward comparison and interpretation, a standardized graphical representation was adopted for all figures. The vertical axis consistently represents the SPA's bending angle, providing a clear and consistent measure of deformation. The horizontal axis denotes the number of bellows, allowing for the assessment of this parameter's effect on bending behavior. Furthermore, to investigate the influence of bellow height, four distinct heights (10 mm, 15 mm, 20 mm, and 25 mm) were simulated and presented as individual lines within each graph: blue for 10 mm, orange for 15 mm, green for 20 mm, and red for 25 mm. This multiparameter analysis, coupled with a standardized graphical representation, provides a comprehensive view of the interaction between key geometric variables and their combined effect on the mechanical response of the SPA structure.

## Experimental test and validation

The experimental test and validation phase is crucial to ensure that the soft-finger actuators meet predefined specifications and perform reliably under operational conditions. This phase involves setting up a detailed experimental system to validate the simulations and theoretical models used during the design phase. The system comprises several key components integrated into a cohesive experimental platform. At the heart of this setup is the pneumatic control system, powered by a pneumatic proportional control valve (SMC VQ110U-5LO) as shown in Fig 16. This valve, which is crucial for precise air pressure regulation, accepts a 24V input voltage and modulates air pressure from 0 to 6 bar, controlled via a voltage range of 0 to

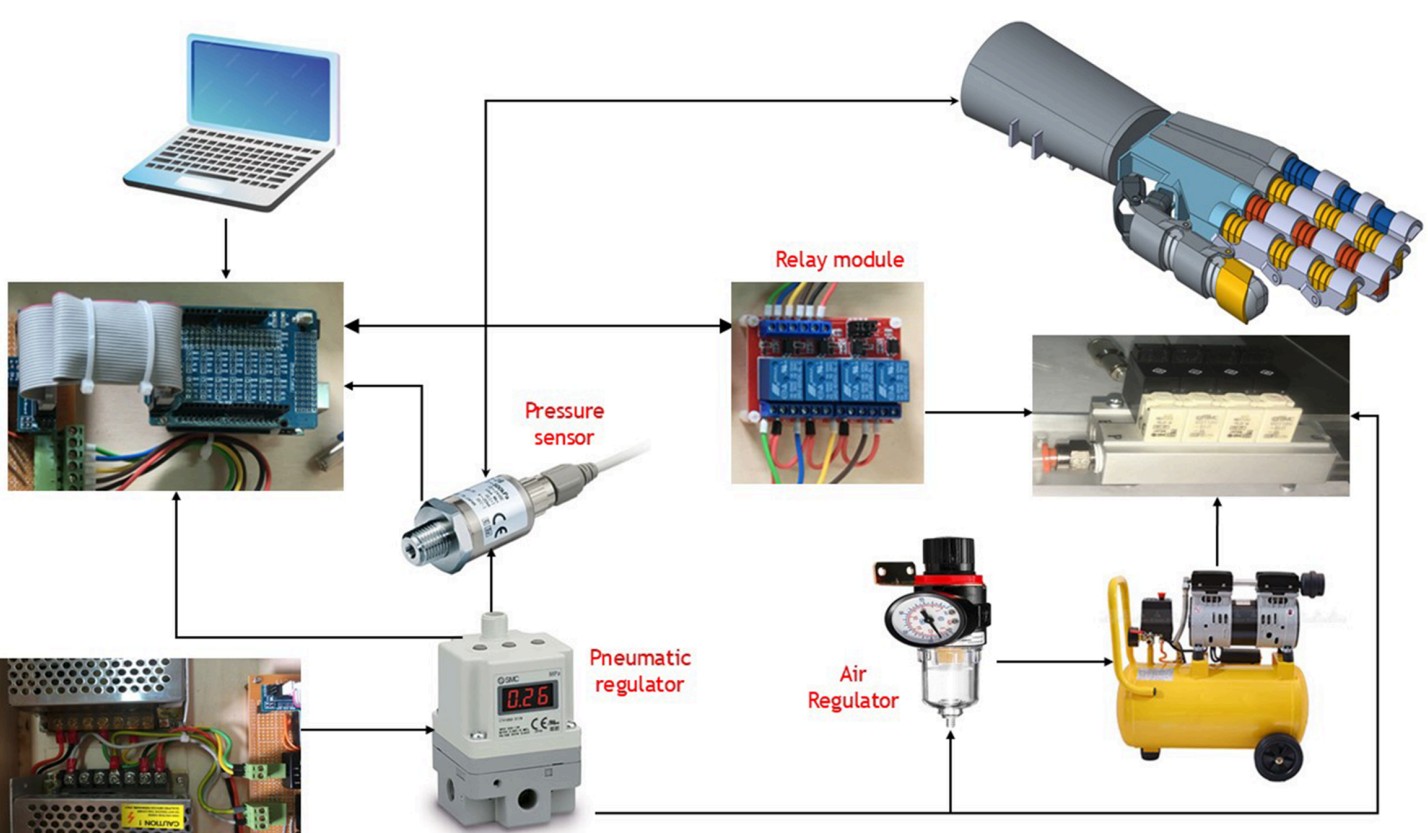

**Fig 16. Schematic Diagram for the pneumatic Circuit, controller and the soft pneumatic hand.**

10V. The controlled air pressure is then directed to the soft finger actuators, passing through an electronic pressure sensor (SMC PSE570-01) that displays real-time pressure readings on a digital interface as shown in Fig 17.

The control signals to the pneumatic valve are generated by an Arduino microcontroller, which interfaces with LabVIEW software configured to produce PWM signals. This setup allows for precise adjustments of the valve operations by varying the PWM duty cycle, which in turn modulates the air pressure output to the actuator as shown in Fig 18. Before starting the experiments, the system undergoes a rigorous calibration process to ensure that each voltage level accurately corresponds to a specific air pressure, thereby standardizing the experimental conditions and enhancing the repeatability of the tests. The experimental platform is specifically designed to accommodate the testing of soft-finger actuators within a controlled environment. This platform includes a mounting setup for actuators, air supply lines, control electronics, and data acquisition systems. The integration of these systems is critical to synchronizing operations and facilitating effective measurement of actuator performance, as shown in Fig 19.

During testing, the actuator's response to different control signals and pressure levels is monitored using high-resolution cameras and specialized tracking software, such as Tracker from Open-Source Physics. This software enables the precise measurement of actuator tip movements in a three-dimensional space, capturing the coordinates x, and y relative to a fixed reference point. This data acquisition is vital for assessing the actuator's mechanical properties, including its bending behavior, response time, and stability under varying loads. After the experiment, the collected data was analyzed to evaluate the actuator's performance against the simulated predictions. This analysis involves comparing the movement trajectories and the bending angles times of the actuator under different experimental conditions, as presented in Figs 20 and 21. Discrepancies between the observed results and the simulations can present the relationship between the design and the manufactured model, which shows good accuracy with low error. The experimental results feed into an iterative design process, where

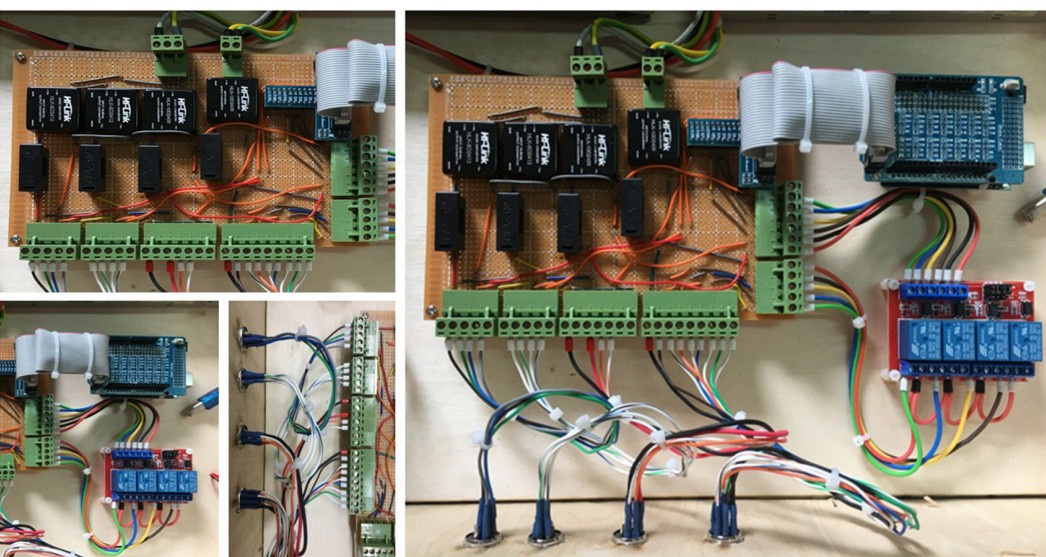

**Fig 17. Control Circuit for the soft pneumatic hand to control the pneumatic system.**

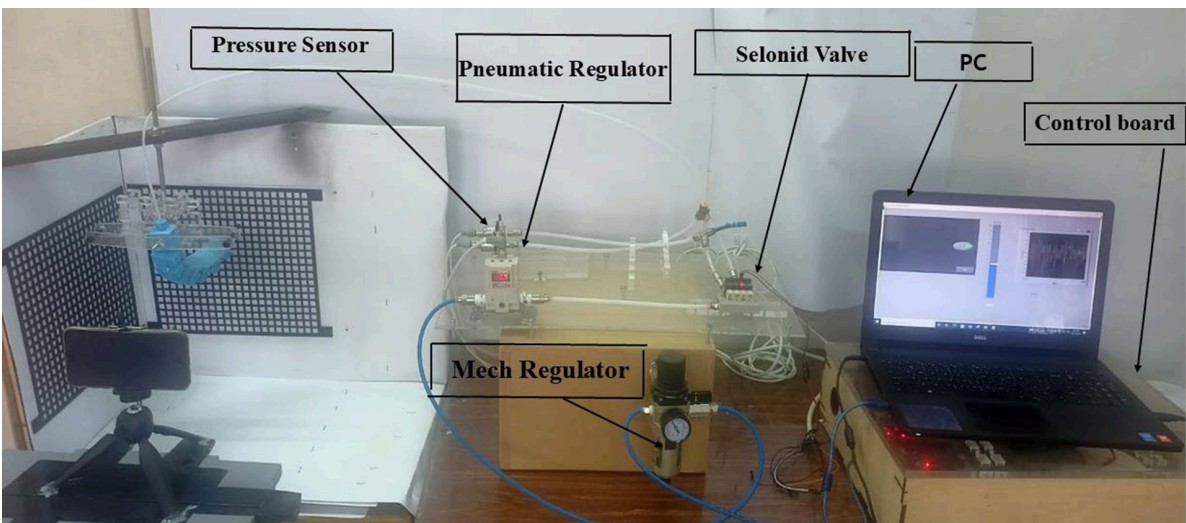

**Fig 18. Overview for the complete test rig, controller, and pneumatic circuit.**

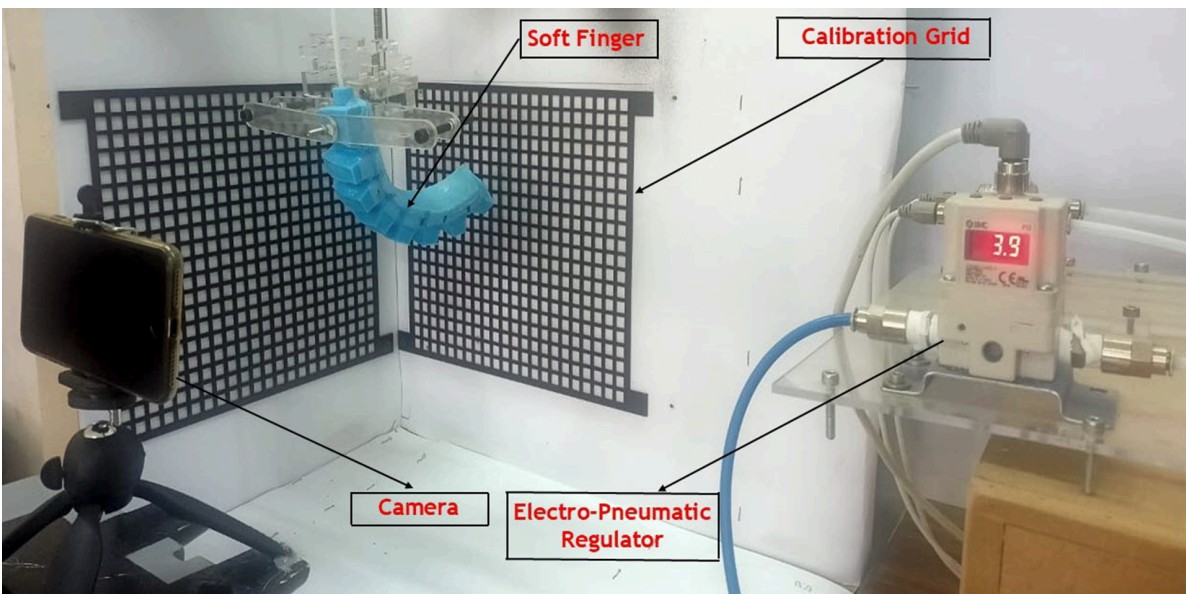

**Fig 19. Test rig during the calibration of the mesh grid to detect the deformation in x and y directions.**

the insights gained from the tests lead to refinements in actuator design. This iterative loop is essential for optimizing the actuator's performance, ensuring that the final product is both efficient and reliable. Adjustments may include tuning control parameters, improving material formulations, or redesigning structural components based on the specific requirements of rehabilitation applications.

Operating Envelope Consideration. While diverse soft actuators can achieve large deformations, we emphasize that our experimental validation targets a sub-100 kPa envelope consistent with rehabilitation safety guidelines and portable pneumatic systems. Under this constraint, the measured bending angles align favorably with or exceed those reported for

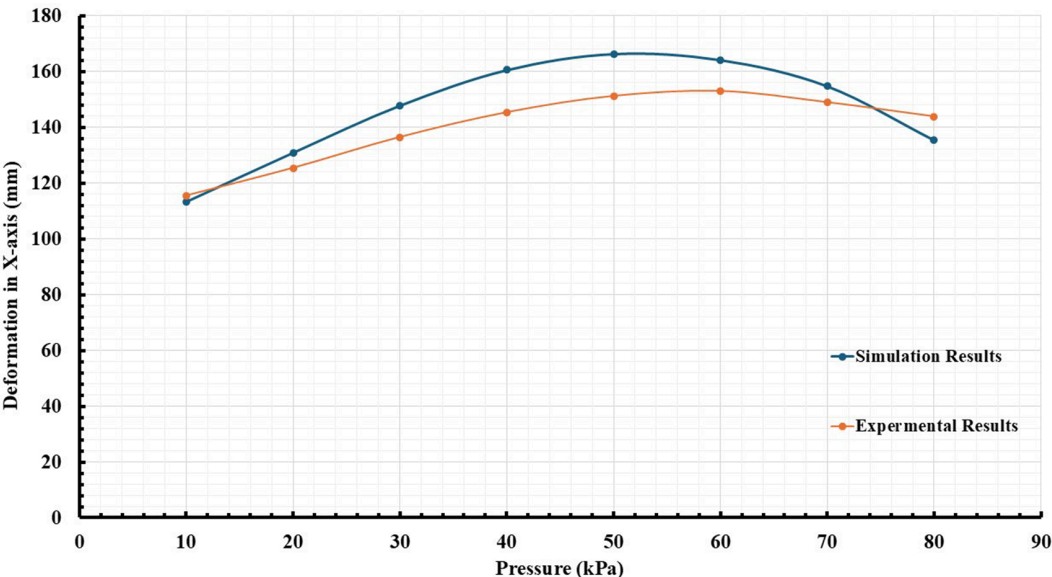

**Fig 20. Comparative analysis of actual and predicted Deformation in the X-axis for the soft actuator performance.**

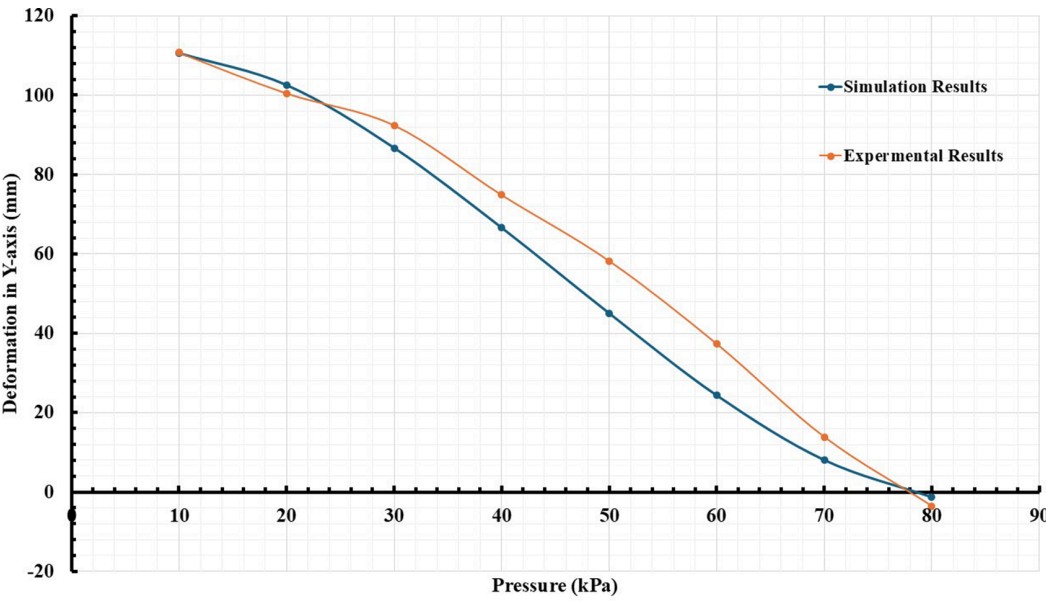

**Fig 21. Comparative analysis of actual and predicted Deformation in the Y-axis for the soft actuator performance.**

comparable chambered designs operated at higher pressures, reinforcing the suitability of the proposed actuator for clinical hand-assist use cases.

## Neural network modeling

This work utilizes an artificial neural network (ANN) to predict the performance of a soft actuator. The ANN aims to minimize the output error while achieving rapid convergence.

The training process involves initializing values and iteratively adjusting them until the network delivers sufficiently accurate results. Initially, the network lacks knowledge and requires training on a dataset of input-output pairs. Fortunately, a comprehensive dataset on the effect of dimensional changes on soft actuator performance is available from previous finite-element simulations. To achieve a more comprehensive model, we designed a network with four input parameters: Height, Number of bellows, Thickness surrounding bellows, and Thickness of the foot. The network generates one output, which is the angle of bending.

Artificial neural networks are computational models inspired by the structure and function of the biological brain [51]. FFNNs are a fundamental type of ANN characterized by a strictly forward direction of information flow, from input to output layers, without loops or cycles [52]. This unidirectional flow distinguishes them from recurrent neural networks (RNNs), which can process sequential data with dependencies. A feedforward neural network (FNN) using TensorFlow/Karas is implemented for regression analysis, exemplifying fundamental principles in machine learning. The initial steps involve data input from an Excel file using pandas, ensuring data integrity and gracefully handling exceptions [53]. The data set is split into training and test sets to assess model generalization, with the FNN architecture comprising dense layers initialized with ReLU activation functions to facilitate nonlinear learning of input features (Number of bellows, Height, Thickness Surrounding Bellow, Thickness of Foot).

Overfitting Mitigation and Validation Protocol. To reduce the risk of overfitting and enhance generalizability, we adopted the following safeguards: (i) a grid-based design-of-experiments (DoE) spanning the four geometric inputs (H, N, TSB, TOF) to ensure broad and uniform coverage of the design space; (ii) stratified splitting into training (70%), validation (15%), and test subsets (15%) preserved the marginal distributions of all inputs; (iii) standardization of the input characteristics (z score) and regularization of the weight of L2 in dense layers; and (iv) early stopping based on validation loss with patience criterion to prevent overtraining. We restricted the model capacity to the smallest architecture that achieved validation convergence, avoiding excessively deep or wide networks that may memorize training samples. To preliminarily assess sim-to-real transfer, we also compared NN predictions with experimental measurements from a set of fabricated actuators not used during model tuning.

Model training employs stochastic gradient descent with the Adam optimizer, minimizing the mean squared error (MSE) loss function to optimize predictions of the target variable (Bending Angle). Over 10,000 epochs and with a batch size of 10, the model learns to map input features to output predictions, leveraging both training and validation datasets for performance evaluation [54,55]. Results are visualized through matplotlib, plotting the trajectory of training and validation losses over epochs to monitor convergence and validate model robustness. Subsequent evaluation on unseen test data quantifies predictive accuracy, demonstrating the FNN's ability to generalize to new observations and its utility in real-world regression tasks.

Fig 22 presents a scatter plot that compares the actual bending angle values (x-axis) to the predicted values generated by our model (y-axis). Ideally, all data points would fall on the diagonal line, indicating perfect prediction, where every predicted value matches the corresponding actual value perfectly. In this plot, the data points exhibit some scatter around the diagonal, indicating a degree of variability in the model's predictions.

A variance analysis was then introduced to evaluate its predictive performance. Initially, the total variability in the target variable 'Bending Angle' was calculated through the total sum of squares (TSS), which measures the squared deviations of each observation from the mean of 'Bending Angle'. Following model predictions on the test set, the residual sum of squares

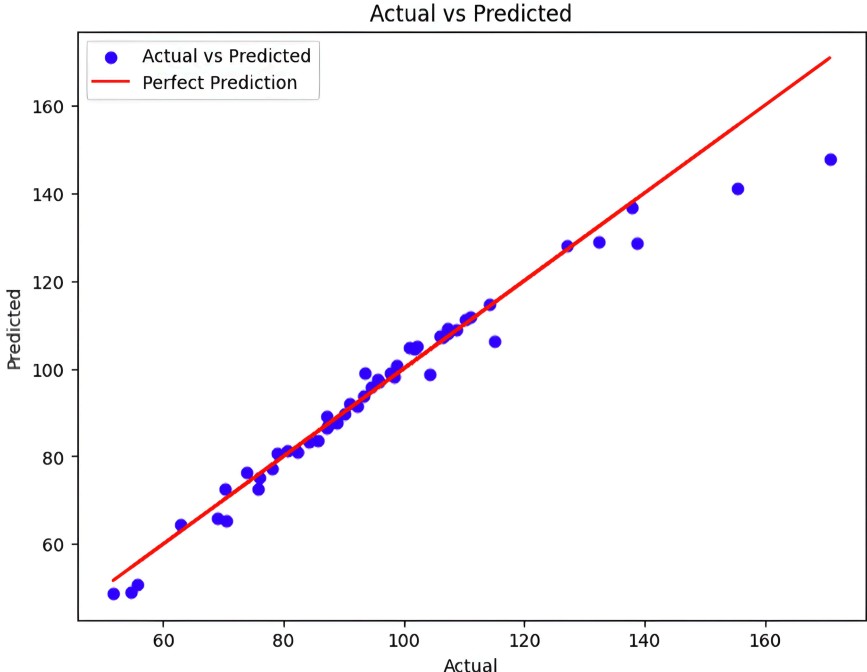

**Fig 22. Performance analysis between predicted and actual data for the actuator pending angle.**

(RSS) quantifies the squared differences between predicted and actual values, assessing how well the model captures observed variance. By computing the percentage of residual variance relative to TSS, the residual variance percentage was determined, indicating the proportion of unexplained variability left by the model. Conversely, the Explained Variance Percentage complements this by highlighting the fraction of variance in the 'Bending Angle' successfully accounted for by the model's predictions. These metrics provide a comprehensive view of the model's efficacy in capturing and explaining data variance, crucial for refining model parameters and enhancing predictive accuracy in practical applications. Evaluation of the data set reveals a remarkably low residual variation percentage of 0. 74%, indicating that the model accounts for the vast majority of the variance present in the data. Consequently, the explained variance percentage stands impressively high at 99.26%, underscoring the model's capacity to accurately predict the 'Bending Angle' based on the provided features. These results affirm the efficacy of the neural network in capturing and utilizing patterns from the training data to make robust predictions on unseen data, highlighting its potential for reliable deployment in practical applications requiring precise regression outcomes. Where Fig 23(a) shows the actual bending angles and Fig 23(b) shows predicted bending angles. Fig 24 presents a comparison between the actual bending angles (blue line) measured from a finger soft actuator and the corresponding predictions generated by a model (green line). The x-axis, labeled "Index," signifies a measurement sequence taken at specific intervals. The y-axis represents the bending angle of the actuator in degrees. The observed and predicted bending angles exhibit a clear correlation, suggesting the ability of the model to capture the overarching trend of the actuator's bending behavior. This indicates good generalizability of the model to the observed actuator dynamics, which indicates its ability to effectively represent the underlying principles governing the bending process.

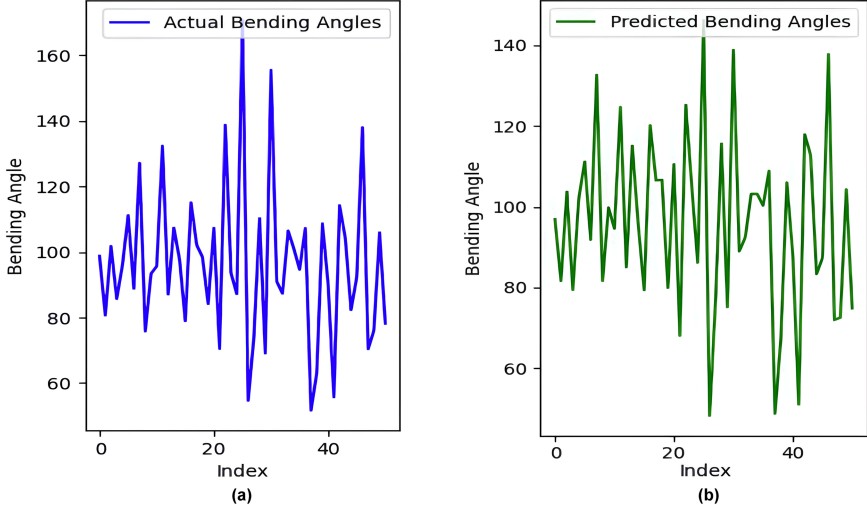

**Fig 23. (a) Actual bending angles; (b) Predicted bending angles.**

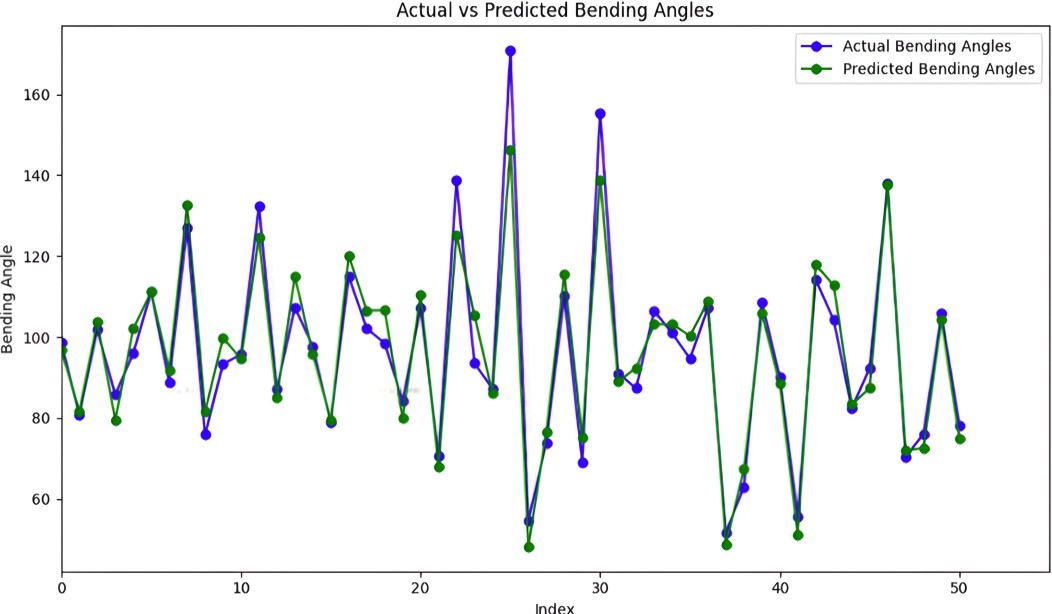

**Fig 24. Comparative analysis of actual and predicted bending angles in soft actuator performance.**

## Conclusions

This study provides valuable insight into optimizing the design of the bellow actuator for various applications by systematically investigating the influence of key geometric parameters on the angle of bending. A clear positive correlation was found between both the height of the bellows and the number of bellows with an achievable bending angle, indicating their importance for applications requiring a greater range of motion. In contrast, increasing the thickness surrounding the bellows (TSB) and the foot thickness resulted in greater system stiffness,

limiting the maximum achievable bending angle despite requiring larger forces for equivalent bending. This highlights a crucial design trade-off between maximizing bending range and maintaining structural integrity, necessitating careful optimization for specific applications. The rigorous methodology of the study, characterized by systematically varying individual parameters while maintaining others constant, allowed the isolation and quantification of their particular contributions to bending behavior, providing a deeper understanding of the underlying mechanics. Furthermore, an artificial neural network (ANN) model, trained on finite element simulation data, demonstrated exceptional accuracy in predicting actuator bending angles based on geometric inputs. This model, with its low residual variance (0.74%) and high explained variance (99.26%), exhibits significant potential for rapid and precise performance prediction, thereby streamlining the design optimization process. Future research will explore the impact of additional geometric parameters and material properties on actuator performance, further enhancing design optimization strategies.

## Supporting information

**TPU material curve. TPU material mechanical properties.** This table presents the relationship between the load and the deformation that has been obtained from a tensile test for a standard Iso specimen test made of TPU.
(PDF)

**Results models at 2 mm. The finite element model for 2 mm.** This table presents the relationship between the load and the deformation on the X-axis, and the Y-axis at different cases for the Yeoh second order, Yeoh third order, Mooney Rivlin (second, Third, and Fifth) orders.
(PDF)

## Author contributions

**Conceptualization:** Mahmoud Elsamanty, Karim Badr, Basem Akl, Hongbo Yang, Kai Guo, Mostafa Orban.

**Data curation:** Mahmoud Elsamanty, Karim Badr, Basem Akl, Abdelkader Ibrahim, Mostafa Orban.

**Formal analysis:** Mahmoud Elsamanty, Karim Badr, Basem Akl, Abdelkader Ibrahim, Mostafa Orban.

**Funding acquisition:** Hongbo Yang, Kai Guo.

**Investigation:** Mahmoud Elsamanty, Karim Badr, Basem Akl, Mostafa Orban.

**Methodology:** Mahmoud Elsamanty, Karim Badr, Mostafa Orban.

**Project administration:** Mahmoud Elsamanty, Hongbo Yang, Kai Guo.

**Resources:** Basem Akl, Hongbo Yang, Kai Guo, Mostafa Orban.

**Software:** Karim Badr, Mostafa Orban.

**Supervision:** Mahmoud Elsamanty, Abdelkader Ibrahim.

**Validation:** Mahmoud Elsamanty, Mostafa Orban.

**Visualization:** Mahmoud Elsamanty, Karim Badr, Mostafa Orban.

**Writing – original draft:** Mahmoud Elsamanty, Karim Badr, Hongbo Yang, Kai Guo, Mostafa Orban.

**Writing – review & editing:** Mahmoud Elsamanty, Mostafa Orban.

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
