## [Decision Letter · Decision Letter 0]

25 Mar 2025

PONE-D-25-03804Design and Optimization of Soft Finger Actuators for Rehabilitation Applications: A Combined Finite Element and Neural Network ApproachPLOS ONE

Dear Dr. Elsamanty,

Thank you for submitting your manuscript to PLOS ONE. After careful consideration, we feel that it has merit but does not fully meet PLOS ONE’s publication criteria as it currently stands. Therefore, we invite you to submit a revised version of the manuscript that addresses the points raised during the review process.

Both reviewers acknowledge the study's significance and methodological rigor but emphasize key revisions: inclusion of comparative analysis with existing technologies, deeper justification for material and ANN choices, clarification of research gaps, detailed explanation of design decisions, and addressing model overfitting. Enhancing discussion on generalizability, experimental limitations, and commercial applicability is also recommended.

We look forward to receiving your revised manuscript.

Kind regards,

Jyotindra Narayan

Academic Editor

PLOS ONE

Journal Requirements:

2. Thank you for stating the following financial disclosure: [This research was funded by the National Key R\\&D Program of China (2023YFB4706200), Chinese Academy of Sciences President’s International Fellowship Initiative (Grant No. 2024VBB0010), Pilot Projects for Fundamental Research in Suzhou (SSD2023014), Science and Technology Development Plan Project of Jilin Province (20240305049YY), Natural Science Foundation Project of Shandong Province (ZR2022QH214), and Chongqing Natural Science Foundation Project (2024NSCQ-MSX0007). Project of the State Administration of Foreign Experts Affairs, H20240225.]. 

3. We note that your Data Availability Statement is currently as follows: [All relevant data are within the manuscript and its Supporting Information files]

Reviewers' comments:

Reviewer's Responses to Questions

**Comments to the Author**

1. Is the manuscript technically sound, and do the data support the conclusions?

Reviewer #1: Yes

Reviewer #2: Yes

2. Has the statistical analysis been performed appropriately and rigorously? 

Reviewer #1: No

Reviewer #2: Yes

3. Have the authors made all data underlying the findings in their manuscript fully available?

Reviewer #1: Yes

Reviewer #2: Yes

4. Is the manuscript presented in an intelligible fashion and written in standard English?

Reviewer #1: Yes

Reviewer #2: Yes

5. Review Comments to the Author

Reviewer #1: This study presents a detailed analysis of soft finger actuators for rehabilitation gloves using finite element modeling and neural networks. It explores how design parameters like bellows number and actuator height affect actuator bending, optimizing performance for therapeutic use. Experimental validation confirms the model's high predictive accuracy. Increasing the bellows height enhances bending angles, while thicker materials decrease flexibility, emphasizing the importance of precise design in developing effective rehabilitation technologies. However, despite its progress, significant concerns remain unaddressed. Some of the raised concerns are mentioedn below:

1. Lack of Comparisons with Existing Technologies: While the manuscript thoroughly investigates the proposed design, it lacks a comparative analysis with existing technologies. Including such comparisons could significantly enhance the relevance and impact of the findings by positioning this research within the larger field of soft robotics.

2. Potential Overfitting in Neural Network Modeling: The high accuracy of the neural network model is impressive; however, the risk of overfitting should be addressed. It would be beneficial for the authors to discuss the steps taken to prevent overfitting and to ensure the model's generalizability to other similar applications.

3. Limited Discussion on Material Choices: The choice of materials, particularly silicone rubber, is briefly discussed in terms of its mechanical properties. However, a more detailed rationale for choosing this material over others, considering the specific requirements of rehabilitation applications, would strengthen the manuscript.

4. Expand on Material Selection Criteria: Provide a more detailed analysis of why silicone rubber was chosen over other potential materials, including an assessment of its performance, durability, and safety in contact with skin.

5. Incorporate Comparative Studies: Add a section comparing the newly designed actuators with existing products or research in the field of soft robotics to highlight the advancements your design provides.

6. Address Model Generalization: Enhance the discussion on the neural network's generalization capabilities by including tests on external datasets or through cross-validation techniques.

7. References: The following refeercnes are highly recommended toa dded to the paper to improve its quality and state of the arts. (10.1007/s41315-025-00421-x), (10.22024/UniKent/01.02.105160), and (10.1109/ACCESS.2023.3325211).

Reviewer #2: The manuscript presents an insightful and well-structured study on the design and optimization of soft finger actuators for rehabilitation applications. The authors employ a combined approach using finite element modeling (FEM) and artificial neural networks (ANN) to investigate and optimize the performance of these actuators. The study meticulously explores key design parameters—such as the number of bellows, actuator height, surrounding thickness, and foot thickness—and their effects on bending behavior. The combination of numerical simulations and experimental validation strengthens the credibility of the findings. The ANN-based predictive modeling demonstrates strong performance, with a high explained variance and low residual error. This research is valuable for rehabilitation applications and contributes significantly to the field of soft robotics. However, there are several areas where the manuscript requires revision to improve clarity, robustness, and scientific impact.

1. The introduction discusses the importance of rehabilitation technologies and soft actuators, but it lacks a clear research gap. The authors should explicitly state what existing limitations in soft actuator design their study addresses.

2. The motivation behind using ANNs should be elaborated. Why was ANN chosen over other modeling techniques? How does it compare to conventional regression models or other machine learning approaches?

3. The choice of thermoplastic polyurethane as the material for the actuator should be justified. Are there any comparisons with other materials in terms of elasticity, fatigue resistance, or biocompatibility?

4. The rationale for the selection of geometric parameters should be further explained. Why were these specific values chosen for the number of bellows, height, and thicknesses? Did preliminary studies influence these choices?

5. More discussion is needed on the limitations of the experimental setup. For instance, does the control system introduce any latency in actuation that could affect real-time performance?

6. The manuscript briefly mentions related work, but it does not explicitly compare the proposed actuator’s performance with existing designs. A comparison table summarizing the advantages and disadvantages of the proposed approach versus previous studies would be beneficial.

7. The ANN model achieves a high explained variance (99.26%), but overfitting concerns should be addressed. Was cross-validation performed? If so, how did the validation set perform compared to the training set?

8. Were alternative architectures tested, such as recurrent neural networks (RNNs) or ensemble learning approaches? If not, a brief justification should be provided.

9. A table summarizing key findings from FEM simulations would provide a clearer reference for the reader.

10. The study presents a well-integrated approach to actuator optimization but could benefit from a more thorough discussion of practical implications. How feasible is the proposed design for commercial rehabilitation devices?

6. PLOS authors have the option to publish the peer review history of their article (what does this mean?). If published, this will include your full peer review and any attached files.

Reviewer #1: No

Reviewer #2: **Yes: **Subhash Pratap

---

## [Author Response · Author response to Decision Letter 1]

2 Sep 2025

All responses to the reviewers’ comments are provided separately, in a dedicated file for each reviewer.

---

## [Decision Letter · Decision Letter 1]

23 Sep 2025

Design and Optimization of Soft Finger Actuators for Rehabilitation Applications: A Combined Finite Element and Neural Network Approach

PONE-D-25-03804R1

Dear Dr. Elsamanty,

We’re pleased to inform you that your manuscript has been judged scientifically suitable for publication and will be formally accepted for publication once it meets all outstanding technical requirements.

Kind regards,

Jyotindra Narayan

Academic Editor

PLOS ONE

Additional Editor Comments (optional):

The reviewers have now recommended the revised manuscript for acceptance. Following the authors' feedback and editor's judgement, the manuscript is being recommended for the publication in its current form. Congratulations to the authors.

Reviewers' comments:

Reviewer's Responses to Questions

**Comments to the Author**

1. If the authors have adequately addressed your comments raised in a previous round of review and you feel that this manuscript is now acceptable for publication, you may indicate that here to bypass the “Comments to the Author” section, enter your conflict of interest statement in the “Confidential to Editor” section, and submit your "Accept" recommendation.

Reviewer #1: All comments have been addressed

Reviewer #2: All comments have been addressed

2. Is the manuscript technically sound, and do the data support the conclusions?

Reviewer #1: Yes

Reviewer #2: Yes

3. Has the statistical analysis been performed appropriately and rigorously? 

Reviewer #1: Yes

Reviewer #2: Yes

4. Have the authors made all data underlying the findings in their manuscript fully available?

Reviewer #1: Yes

Reviewer #2: Yes

5. Is the manuscript presented in an intelligible fashion and written in standard English?

Reviewer #1: Yes

Reviewer #2: Yes

6. Review Comments to the Author

Reviewer #1: The revised paper has been improved significantly, addressing the major concerns raised in the initial review. The authors have strengthened the methodology, clarified the presentation of results, and enhanced the overall discussion of the findings. These revisions greatly improve the clarity, scientific rigor, and impact of the work. The manuscript now represents a valuable contribution to the literature, and I strongly recommend it for publication.

Reviewer #2: (No Response)

7. PLOS authors have the option to publish the peer review history of their article (what does this mean?). If published, this will include your full peer review and any attached files.

Reviewer #1: No

Reviewer #2: **Yes: **SUBHASH PRATAP

---

## [Editor Report · Acceptance letter]

PONE-D-25-03804R1

PLOS ONE

Dear Dr. Elsamanty,

I'm pleased to inform you that your manuscript has been deemed suitable for publication in PLOS ONE. Congratulations! Your manuscript is now being handed over to our production team.

Kind regards,

on behalf of

Dr. Jyotindra Narayan

Academic Editor

PLOS ONE